# Type I Diabetes Mellitus Suppresses Experimental Skin Carcinogenesis

**DOI:** 10.3390/cancers16081507

**Published:** 2024-04-15

**Authors:** Maria Giakoumaki, George I. Lambrou, Dimitrios Vlachodimitropoulos, Anna Tagka, Andreas Vitsos, Maria Kyriazi, Aggeliki Dimakopoulou, Vasiliki Anagnostou, Marina Karasmani, Heleni Deli, Andreas Grigoropoulos, Evangelos Karalis, Michail Christou Rallis, Homer S. Black

**Affiliations:** 1Division of Pharmaceutical Technology, Department of Pharmacy, School of Health Sciences, National and Kapodistrian University of Athens, Panepistimiopolis, 15784 Athens, Greece; giakoumaki.mar@gmail.com (M.G.); avitsos@pharm.uoa.gr (A.V.); kyriazimaria@mail.ntua.gr (M.K.); aggelikidimakopoulou1991@gmail.com (A.D.); vassia.an@gmail.com (V.A.); marinakarasmani44@gmail.com (M.K.); ed1990@windowslive.com (H.D.); a.grigoropoulos@famar-group.com (A.G.); vkaralis@pharm.uoa.gr (E.K.); 2Choremeio Research Laboratory, First Department of Pediatrics, School of Health Sciences, Medical School, National and Kapodistrian University of Athens, Thivon & Levaeias 8, Goudi, 11527 Athens, Greece; glamprou@med.uoa.gr; 3Research Institute of Maternal and Child Health & Precision Medicine, National and Kapodistrian University of Athens, Thivon & Levadeias 8, 11527 Athens, Greece; 4Department of Forensic Medicine and Toxicology, Medical School, National and Kapodistrian University of Athens, 75, Mikras Asias Street, 11527 Athens, Greece; dvlacho@med.uoa.gr; 5First Department of Dermatology and Venereology, ‘Andreas Syggros” Hospital, School of Medicine, National and Kapodistrian University of Athens, Ionos Dragoumi 5, 11621 Athens, Greece; annatagka3@gmail.com; 6Department of Dermatology, Baylor College of Medicine, Houston, TX 77030, USA; hblack@bcm.edu

**Keywords:** Diabetes Mellitus, mouse model, carcinogenesis, ultraviolet, squamous cell carcinoma, pigmented nevi

## Abstract

**Simple Summary:**

This research investigates the novel area of how Type 1 and Type 2 diabetic skin responds to ultraviolet (UV) radiation, compared to normal skin, a subject previously unexplored. By focusing on the well-documented carcinogenic effects of UV radiation on murine skin, the study compares these effects with those on diabetic murine skin. For the first time, our findings reveal that Type 1 diabetic skin shows reduced sensitivity in developing squamous cell carcinoma and nevi. This research could significantly impact the scientific community by enhancing our understanding of skin cancer’s pathogenesis in diabetic mice and could potentially guide future research on skin carcinogenesis.

**Abstract:**

This study explores the previously uncharted territory of the effects of ultraviolet (UV) radiation on diabetic skin, compared to its well-documented impact on normal skin, particularly focusing on carcinogenesis and aging. Employing hairless SKH-hr2, Type 1 and 2 diabetic, and nondiabetic male mice, the research subjected these to UV radiation thrice weekly for eight months. The investigation included comprehensive assessments of photoaging and photocarcinogenesis in diabetic versus normal skin, measuring factors such as hydration, trans-epidermal water loss, elasticity, skin thickness, melanin, sebum content, stratum corneum exfoliation and body weight, alongside photo documentation. Additionally, oxidative stress and the presence of hydrophilic antioxidants (uric acid and glutathione) in the stratum corneum were evaluated. Histopathological examination post-sacrifice provided insights into the morphological changes. Findings reveal that under UV exposure, Type 1 diabetic skin showed heightened dehydration, thinning, and signs of accelerated aging. Remarkably, Type 1 diabetic mice did not develop squamous cell carcinoma or pigmented nevi, contrary to normal and Type 2 diabetic skin. This unexpected resistance to UV-induced skin cancers in Type 1 diabetic skin prompts a crucial need for further research to uncover the underlying mechanisms providing this resistance.

## 1. Introduction

Skin serves as a critical interface between the environment and the body, providing essential barrier functions against various environmental pollutants, including chemical and physical agents, as well as ultraviolet (UV) light [1,2]. The overexposure to UV radiation leads to the production of a significant amount of Reactive Oxygen Species (ROS), overwhelming the tissue’s antioxidant defenses and activating pathways dependent on oxidants. Such unregulated ROS release plays a key role in the development of numerous skin disorders, including cutaneous neoplasia [3]. UV radiation inflicts both acute and chronic damage to the skin [4], where acute effects manifest as severe inflammation, and chronic exposure results in ageing and cancer [5]. Specifically, UVB radiation causes damage to the DNA of keratinocytes, disrupting the normal keratinization process with chronic exposure and leading to proliferative diseases, such as basal and squamous cell carcinoma [5]. Although UVA is less carcinogenic than UVB, it also contributes to photo-carcinogenesis, primarily through the generation of ROS that induce DNA lesions [6].

Skin tumors are notably prevalent within the Caucasian population [7]. These tumors are categorized into Melanoma Skin Cancer (MSC) and non-Melanoma Skin Cancer (NMSC), with the latter encompassing Basal Cell Carcinoma (BCC) and Squamous Cell Carcinoma (SCC) types. MSC represents the most aggressive form of skin tumor, whereas NMSCs are generally associated with a better prognosis, despite being more prevalent but of lower malignancy grade. Among NMSCs, SCC is identified as the most aggressive, with a higher propensity for tissue invasion [8,9,10].

Diabetes Mellitus (DM) is defined as a chronic, systemic condition characterized by hyperglycemia, leading to severe complications, including neuropathy [11]. The skin is frequently affected by DM, with a significant portion of diabetic patients experiencing skin-related complications [12]. Xerosis, an early and common skin abnormality among diabetics, is believed to significantly contribute to skin fissures, hyperkeratosis, and ulceration [13,14]. DM is recognized as a major public health issue, responsible for numerous micro- and macro-vascular diseases. The complications arising from DM are largely attributed to chronic hyperglycemia and elevated fatty acid levels, with oxidative stress playing a crucial role due to the increased production of free radicals and their diminished degradation by cellular antioxidant mechanisms [14]. This oxidative stress is closely linked to high glucose levels, the generation of free radicals, and the onset of diabetic complications.

Dermatological issues are reported in approximately one-third of diabetic patients [13]. Chronic ulcers, resulting from incomplete healing, represent the most prevalent skin disorder, potentially leading to infections and limb amputation [15]. Other common skin conditions include lipoid necrosis, Acanthosis Nigricans (AN), sclerosis, and Granuloma Annulare (GA) [16]. Factors such as generalized itching, dry skin, delayed healing, and skin infections, significantly impact the quality of life of diabetic patients, contributing to various skin diseases [17]. Dry skin, in particular, is one of the earliest symptoms of skin abnormality in diabetic patients, playing a critical role in the development of skin cracks, hyperkeratosis, and diabetic foot ulcers [18]. Currently, there is a lack of epidemiological data linking diabetes to photo-carcinogenesis, as well as a dearth of experimental studies examining the relationship between diabetes and non-melanoma skin carcinogenesis.

The present study aims to explore the correlation between UV radiation and diabetic skin, specifically focusing on the susceptibility of squamous cell carcinoma in UV-exposed diabetic skin, utilizing the SKH-hr2 hairless mice model as the basis for investigation.

## 2. Materials and Methods

### 2.1. Animals

The study utilized 42 male SKH-HR2 mice, aged 2 months, maintained in a controlled environment within the Small Animal Laboratory at the National and Kapodistrian University of Athens, School of Pharmacy (Reg. EL 25 BIO-BR 06). Conditions were set to 23 ± 1 °C, with 25–45% humidity, under yellow fluorescent tubes following a 12-h light–dark cycle. The experimental procedures adhered to the animal care guidelines of the European Council Directive 2010/63/EU. The mice had unlimited access to a standard chow diet and fresh water and were acclimatized to these conditions for a week before the commencement of experiments, before being divided into six groups of seven mice each. All procedures were performed in accordance with ARRIVE guidelines [19].

### 2.2. UV Radiation/Dosimetry

Solar simulated UV radiation (280–400 nm) was obtained from a 1000 W Xenon lamp with a UVA power of 6.5 mW/cm^2^ and UVB power of 6.5 mW/cm^2^, before being placed in an Arc Lamp Housing (66020-M), and connected with a Universal Power Supply (68820) from Oriel Instruments (Stratford, CT, USA). The light was filtered through appropriate filters for these wavelengths. The irradiance (after filtering) was measured before every experiment by a Goldilux Smart Meter (70239) from Oriel Instruments (USA). The dose, measured in Minimal Erythemal Dose (MED), was 0.75 MED (4.5 s) in the first week and increased by 25% a week until stabilization at 3 MED (18 s). The Minimal Erythemal Dose (MED) refers to the lowest dose of ultraviolet (UV) radiation that will produce a noticeable redness or erythema (sunburn) on the animal’s skin.

### 2.3. Induction of Diabetes Mellitus

Mice were treated with two different streptozocin doses. Type 2 diabetes, referred to throughout the manuscript, was induced by five daily intraperitoneal injections of 20 mg/kg of streptozocin, low dose in sodium citrate buffer 0.1 M, pH 3.5–4.5 (Zanosar, Pharmacia, Upjohn, Kalamazoo, MI, USA). This treatment produced a mild Diabetes Mellitus response (Diabetes Type 2). Full-blown Diabetes Mellitus Type 1, was induced in mice by five daily intraperitoneal injections of 30 mg/kg of streptozocin high dose in sodium citrate buffer 0.1 M, pH 3.5–4.5 (Zanosar, Pharmacia, Upjohn, USA). The glucose content was measured with a glucose striping method (ABBOT Freestyle Precision, Alameda, CA, USA), according to the manufacturer’s instruction.

### 2.4. Experimental Design

The mice were divided into six groups (Group I–VI). Each group consisted of seven mice (*n* = 7). Group I consisted of the reference group, which did not receive any treatment, neither for diabetes nor for UV irradiation. Group II included mice treated with 20 mg/kg streptozocin (Type 2 diabetes) but no UV irradiation. Group III included mice treated with 30 mg/kg streptozocin (Type 1 diabetes) but no UV irradiation. Group IV included mice, which received no streptozocin treatment but received UV irradiation. Group V included mice treated with 20 mg/kg streptozocin (Type 2 diabetes) and UV irradiation. Group VI included mice treated with 30 mg/kg streptozocin (Type 1 diabetes) and UV irradiation. Animals were weighed during the experiment. The experimental design is summarized in Table 1 and Figure 1.

### 2.5. Photo-Documentation

The skin was evaluated with photo documentation via a Nikon photo camera, Nikkor AF-S Micro 60 mm f/2.8 G ED, SWMED IF Aspherical camera (Tokyo, Japan). The distance between the camera and the skin was 30 cm.

### 2.6. Measurement of Stratum Corneum Trans-Epidermal Water Loss (TEWL)

TEWL was measured using a TEWAMETER^®^ TM 240 (Courage-Khazaka, Köln, Germany), each month. TEWL can be considered as an indicator of the integrity of the epidermal water diffusion barrier and was expressed in gr/m^2^/h.

### 2.7. Measurement of Melanin Content Pigmentation

Melanin was measured using MEXAMETER^®^ MX 18 (Courage-Khazaka, Germany), each month. This device is a well-established narrow-band absorbance/reflectance meter used for assessing the quantity of the major component (melanin) responsible for skin color. The results are shown in arbitrary units (a.u.).

### 2.8. Measurement of Stratum Corneum Hydration

The state of Stratum Corneum hydration was determined each month using CORNEOMETER^®^ CM 820 (Courage-Khazaka, Germany). The results are shown in arbitrary units (a.u.).

### 2.9. Assessment of Elasticity

The skin mechanical properties were measured using CUTOMETER^®^ (Courage-Khazaka, Germany), each month. The measuring principle is based on the suction method. This measurement principle makes it possible to obtain information about the elastic and mechanical properties of skin surface and objectively quantify skin aging. The maximum depth (mm) of skin penetration into the probe when the vacuum is applied is termed R0: this factor measures “skin distensibility and reflects skin firmness” [20]. The ability of the skin to return to its original state after the vacuum is released is termed *R*2: it reflects “gross skin elasticity” and is calculated as: *R*2 = *U*_a_/*U*_f_. *U*_f_ is the maximal distension of the skin into the probe at the end of the vacuum period (i.e., U Rf = 0) And Ua is the total recovery of the skin toward its original position after one second of normal pressure [20]. Higher R0 values indicate skin that has greater capacity to deform, while higher R2 values indicate greater elasticity. The results are shown in arbitrary units (a.u.).

It is important to note that the term “elasticity”, as used in the context of Cutometer measurements and reported in a vast body of literature [20], encompasses both elastic and viscoelastic properties of the skin. The Cutometer has been designed to quantify the viscoelastic properties of the skin by measuring its ability to resist and recover from deformation, which is an indicator of its elasticity but is not a direct measurement of elasticity.

### 2.10. Assessment of Skin Thickness

Since UV radiation confers hyperkeratosis effects to skin [21], we have evaluated the skinfold thickness with a digital caliper (Powerfix Profi, Milomex Ltd., Bedfordshire, UK). It is noteworthy that skin fold measurement is not a direct method for assessing skin thickness in mice; however, with careful application and interpretation, it can provide insights into changes in skin and subcutaneous tissue properties in a cost effective and non-invasive manner [22].

### 2.11. Measurement of Sebum

The sebum of the skin surface was determined by using a Sebumeter^®^ (Courage-Khazaka, Germany).

### 2.12. Hydrophilic Antioxidant Molecules Extraction and Measurement

Adhesive tapes, each measuring 2 × 2 cm, were pressed three times onto the backs of the irradiated mice. The weight of each tape was accurately measured before and after use using a precision scale (Mettler-Toledo, Greifensee, Switzerland), aiming to quantify the Stratum Corneum’s hydrophilic antioxidant molecules. These tapes were then placed into 1.5 mL Eppendorf tubes containing 0.5 mL of HPLC grade water (ThermoFisher Scientific, Waltham, MA, USA), a mixture of Methanol: EDTA (90:10) (Sigma-Aldrich GmbH, Munich, Germany), Butylated Hydroxytoluene (BHT), and Deferoxamine (Sigma-Aldrich GmbH, Germany). The vials underwent vortexing at 2500 rpm for one minute and were subsequently centrifuged at 4000 rpm at 5 °C for seven minutes (Sigma 202 MK). The solvent extracted from this process was stored at −80 °C until it was further processed. The levels of Glutathione and Uric acid were determined using High Performance Liquid Chromatography (HPLC) with electrochemical detection, following a validated method, as previously described [23].

### 2.13. Measurement of Oxidative Stress in Stratum Corneum

Reactive Oxygen Species (ROS) levels were measured using CM-H2DCFDA, a chloromethyl derivative of 2′,7′-dichlorodihydrofluorescein diacetate (H2DCFDA). This compound is a cell-permeant indicator that passively diffuses into cells. Inside the cells, its acetate groups are removed by intracellular esterases, and its thiol-reactive chloromethyl group interacts with intracellular glutathione and other thiols. This interaction, followed by oxidation, results in the formation of a fluorescent adduct that remains trapped inside the cell, thereby enabling long-term studies [24]. For the experiment, 50 μg of CM-H2DCFDA was diluted in 600 µL of ethanol (Molecular Biology Grade (MBG), Sigma-Aldrich GmbH, Germany). Then, 7 μL of this solution was added to 200 μL of the stripped cells solution in each well of a 96-well plate (ThermoFisher Scientific, MA, USA). The first and last wells were designated as blanks, filled with 200 µL of NaCl solution and fluorescent dye (Fluorescein), respectively. The plate was promptly placed in the Fluostar plate reader (BMG Labtech, Ortenberg, Germany) for ROS measurement, with detection set at an excitation wavelength of 485 nm and emission wavelength of 520 nm.

### 2.14. Statistical and Data Analysis

#### 2.14.1. Statistical Analysis

All measures were performed at least in triplicate. All results are expressed as the mean ± SD (Standard Deviation). Normal distribution was evaluated with a Kolmogorov–Smirnov test. Groups were compared by unpaired t-tests, analysis of variance (ANOVA), followed by Bonferroni’s and Fisher’s Least Significant Difference (LSD) multiple comparison tests. All statistical analyses were performed using SPSS statistics 17.00 software, with *p* < 0.05 defined as statistically significant. 

Chi-square test of independence was used to evaluate the association between samples’ characteristics. The characteristics that were found statistically significant were entered in a logistic regression model to evaluate the probability of having multiple positive reactions. The modeling of a quantitative variable based on one or more qualitative and quantitative parameter was performed through linear regression. Multiple logistic regression was performed to evaluate the probability of having multiple positive reactions. The Relative Risk (RR), Odds Ratio (OR), Absolute Risk (AR) were calculated. 

#### 2.14.2. Data Analysis

##### Hierarchical Clustering and K-Means

Data were further analyzed for common expression patterns using classification methods (Matlab^®^ software 2023B) To gain further insight, unsupervised Hierarchical Clustering (HCL) and k-means classification [25,26] was used. HCL with dendrogram was used, and correlations were calculated with Euclidean Distance. K-means classification [25,26] was recently reported as one of the best performing clustering approaches for microarray class discovery studies [27]. The squared Euclidean was applied as a distance measure, as it is generally considered to be a more appropriate measure for use with k-means and has been found to outperform ratio-based measurements [28]. A total of 100 iterations were used and the optimal cluster number for the k-means algorithm was estimated using the Calinski–Harabasz criterion. Complete k-means clusters, centroids, and sorted centroids [29] were utilized.

##### Receiver Operating Characteristics (ROC) Analysis

ROC curves and Naïve Bayes Classification were used to investigate the diagnostic ability of the evaluated parameters. In the case of Naïve Bayes classification, the algorithm, which uses the Bayes theorem and (naively) assumes that the predictors are conditionally independent, is given a class. Naive Bayes classifiers assign observations to the most probable class (in other words, the maximum a posteriori decision rule).

## 3. Results

### 3.1. Descriptive Statistics of our Measurements

#### 3.1.1. Glucose

The initial approach included the evaluation of glucose levels in the experimental mouse model (Table 2). Diabetes Mellitus Type 1 induction has been successful, as both groups (diabetic (III), as well as irradiated Type 1 diabetic (VI) mice, manifested higher glucose levels, as compared to control mice. Glucose levels increased from ~200 mg/dL at the beginning of the experiment to ~400 mg/dL, and then remained relatively stable at ~500 mg/dL (Table 2 and Figure 2A). These differences were significant in the diabetic mice (Group III) and the irradiated diabetic mice (Group VI). Yet, the Type 2 diabetic mice (both irradiated and non-irradiated) manifested marginal glucose increased levels (Table 2 and Figure 2A). When comparing the initial- and end-phase of the experiment (that is at month 0 and month 6), significant differences were observed for the Type 2 diabetic group (Group II), the diabetic group (Group III) and the irradiated diabetic group (Group VI), yet not for the irradiated Type 2 diabetic group (Group V) (Table 2 and Figure 2B,C). However, comparison of the irradiated Type 2 diabetic group at other time points, like at month 5, showed significant differences were obtained, compared to initial (month 0) glucose levels (Table 2).

#### 3.1.2. Trans-epidermal Water Loss (TEWL)

Interestingly, TEWL was lower in the diabetic (Group II and III) and control mice (Group I), as compared to the irradiated mice (Groups IV and V) (Figure 3A). The interesting finding was that irradiated Type 1 diabetic mice (Group VI) manifested significantly lower TEWL levels, as compared to irradiated control mice (Group IV) and irradiated Type 2 diabetic mice (Group V) (Figure 3A). In addition, irradiated (Group IV) and irradiated Type 2 diabetic (Group V) mice manifested significantly higher levels of TEWL between months 0 and 6 (Figure 3B). Similarly, irradiated Type 1 diabetic mice (Group VI) manifested significantly higher TEWL levels between months 0 and 6 (Figure 3B). Finally, it appeared that diabetes rescued the irradiated mice from high TEWL levels, as compared to the other two irradiated mouse groups at month 6 (Figure 3C). Further, TEWL levels were similar when compared to the non-irradiated mice (Figure 3C). The TEWL measurements are summarized in Table 3.

#### 3.1.3. Hydration

Hydration levels can be distinguished into three possible classes (Figure 4A). The control group (Group I) and Type 2 diabetic group (Group II) manifested similar levels of hydration, which remained relatively constant for the duration of 6 months. The second class consisted of the Type 1 diabetic mice (Group III), the irradiated control mice (Group IV) and the irradiated Type 2 diabetic mice (Group V), which manifested a slight decrease in hydration levels. The third one consisted of the irradiated Type 1 diabetic group (Group VI), which manifested a significant decrease in the hydration levels, as compared to all other groups (Figure 4A). When comparing the hydration levels between month 0 and month 6, it appeared that Type 1 diabetic mice (Group III) and all irradiated groups (IV, V, VI) manifested significantly lower hydration levels (Figure 4B). Interestingly, control mice (Group I) and Type 2 diabetic (Group II) as well as irradiated control mice (Group IV) and irradiated Type 2 diabetic (Group V) manifested similar levels of hydration at month 6 (Figure 4C). The Type 1 diabetic irradiated and non-mice manifested significantly different levels of hydration among the experimental groups (Figure 4C). The hydration levels measurements are summarized in Table 4.

#### 3.1.4. Skin Layer Thickness Levels

Skin thickness manifested an ascending tendency, with significant differences in the irradiated control group (Group IV) and the irradiated Type 2 diabetic group (Group V) (Figure 5A). Interestingly, the control group (Group I), the Type 2 diabetic group (Group II) and the Type 1 diabetic group (Group III) manifested no significant change, as compared to month 0 as well as all groups, they were found to have similar skin thickness (Figure 5A). A slight increase in skin thickness was found in the irradiated Type 1 diabetic group (Group VI), yet this increase was close to the non-irradiated experimental groups (Figure 5A). When comparing the skin thickness with respect to month 0 and month 6, all experimental groups manifested significantly higher levels at month 6, as compared to month 0 where, with the exception of Group III, Type 1 diabetic mice manifested lower levels of skin thickness at month 6 (Figure 5B). In comparing the experimental groups, we have found that no significant difference was observed between the control group (Group I) and the Type 2 diabetic group (Group II), while the irradiated Type 1 diabetic mice (Group VI) manifested significantly lower skin thickness levels, as compared to the other two irradiated mice groups (Figure 5C). In addition, the irradiated mice groups manifested higher thickness levels, as compared to the non-irradiated mice groups (Figure 5C). The skin layer thickness measurements are summarized in Table 5.

#### 3.1.5. Body Weight

Body weight is a significant factor in the evaluation of diabetes. Irradiated Type 1 diabetic mice (Group VI) manifested significant weight loss, as compared to the other groups (Figure 6A). All other groups manifested an increase in body weight, with respect to time (Figure 6A). This was also apparent in the comparison of month 0 and month 6, where all groups manifested an increase in body weight, with the exception of Type 1 diabetic mice (Group III), which remained constant, and irradiated Type 1 diabetic mice (Group VI), which lost weight (Figure 6B). Differences were also significant in all groups at month 6 (Figure 6C). Irradiated Type 1 diabetic mice manifested the greatest weight loss, as compared to all other groups (Figure 6C). The body weight measurements are summarized in Table 6.

#### 3.1.6. Skin Elasticity

Skin elasticity manifested two clusters of measurements (Figure 7A). It appeared that control (Group I), Type 2 diabetic (Group II) and Type 1 diabetic (Group III) mice manifested similar elasticity levels, while control irradiated (Group IV), irradiated Type 2 diabetic (Group V) and irradiated Type 1 diabetic (Group VI) mice also manifested similar elasticity levels (Figure 7A). It appeared that the UV radiation was the separating factor in skin elasticity. In the non-irradiated mice, the Type 2 diabetic mice manifested significantly higher levels of skin elasticity (Figure 7B), while the irradiated groups (Group IV, V, VI) manifested significantly lower levels of elasticity at month 5, as compared to month 0 (Figure 7B). Finally, elasticity was found to be significantly higher in the non-irradiated group, as compared to the irradiated group (Figure 7C). The skin elasticity measurements are summarized in Table 7.

#### 3.1.7. Sebum, Melanin, and Stripped Keratinocytes

Sebum measurements showed that the control irradiated group (Group IV) manifested significantly higher levels of sebum, as compared to all other groups (Figure 8A). It appeared that sebum levels were correlated to the presence of diabetes. Similarly, melanin manifested significantly higher levels in the irradiated groups, as compared to the non-irradiated groups (Figure 8B). The irradiated groups (groups IV, V, VI) manifested higher levels of keratinocyte loss, as compared to the non-irradiated groups (Figure 8C). The sebum, melanin and stripped keratinocyte measurements are summarized in Table 8.

#### 3.1.8. Uric Acid

Uric acid was evaluated during months 3 and 5 in the experimental mice model. In particular, diabetic mice (Group III) manifested significantly higher uric acid levels at month 5, as compared to month 3 (Figure 9A). In addition, irradiated diabetic mice manifested significantly lower levels of uric acid at month 5, as compared to month 3 (Figure 9B). On the other hand, no significant differences were observed at month 5 among all investigated groups (Figure 9B). At the same time, control irradiated (Group IV) and irradiated diabetic (Group VI) mice manifested significantly higher uric acid levels, as compared to non-irradiated groups (Figure 9B). The uric acid measurements are summarized in Table 9.

#### 3.1.9. Glutathione

Glutathione was evaluated during months three and five (as in the case of uric acid) in the experimental mice model. Irradiated Type 1 diabetic and control mice (Groups IV and VI) manifested significantly higher glutathione levels in the third month, as compared to the fifth month (Figure 10A). Control mice (Group I) manifested higher levels of glutathione, as compared to Type 1 diabetic (Group III), and irradiated control and Type 1 diabetic mice (Groups IV and VI) at both tested times (Figure 10B). In addition, Type 1 diabetic mice (Group III) manifested significantly higher glutathione levels, as compared to irradiated Type 1 diabetic (Group VI) mice at both tested times (Figure 10B). The glutathione measurements are summarized in Table 9.

#### 3.1.10. Oxidative Stress

Oxidative stress was measured via the presence of reactive oxygen species using fluorescence. No significant results were found, except that control irradiated (Group IV) mice manifested significantly higher ROS levels, as compared to control (Group I) mice (Figure 11). It appeared that the decisive inhibiting factor for the presence of ROS was diabetes. The uric acid measurements are summarized in Table 9.

### 3.2. Dermatoscopic Evaluations

Beside the parameters evaluated, the effects of UV on the experimental groups were also assessed, using dermatoscopy. Irradiated control mice (Group IV) manifested papilloma and hyperkeratosis after six months of UV exposure (Figure 12A–C). On the other hand, irradiated Type 2 diabetic mice (Group V) also manifested papilloma and hyperkeratosis, yet to a lesser extent, as compared to the control group (Figure 12D–F). On the contrary, irradiated Type 1 diabetic mice (Group VI) did not manifest any papilloma or hyperkeratosis, indicating that Type 1 diabetes functioned as a protective factor for mice skin (Figure 12G–I).

### 3.3. Photographic Evaluations

The mice were also examined macroscopically, to observe for skin benign or malignant aberrations. Non-irradiated mice (including control mice (Group I)), Type 2 diabetic (Group II) and Type 1 diabetic mice did not manifest any serious skin aberrations during the complete course of the experiment (Figure 13). Type 1 diabetes only manifest skin dryness (Figure 4, Figure 5, Figure 8 and Figure 13). As expected, in the control irradiated group (Group IV), significant skin aberrations were manifested after six months of UV exposure, i.e., the evident emergence of SCC, as well as in the irradiated Type 2 diabetic group (Group V), where milder SCC skin transformations were observed (Figure 14). On the contrary, in the irradiated Type 1 diabetic group (Group VI), no skin transformations were identified, and only an extended melanosis was observed (Figure 14).

### 3.4. Histological Evaluation of the Experimental Model

Histological evaluations of the study mice confirmed the previously reported results. Control animals (Group I) manifested no cytological transformations (Figure 15A), as well as those animals in Group II (Figure 15B) and Group III (Figure 15C). On the other hand, mice in Group IV manifested squamous cell carcinoma (SCC), which was presented as a high-grade, well-differentiated and non-invasive SCC (Figure 15D), high-grade, well-differentiated, invasive SCC (Figure 15E) and high-grade, moderately differentiated, invasive SCC (Figure 15F). Similar results were obtained from animals in Group V, where mice were presented with high-grade, well-differentiated, invasive SCC (Figure 15G), high-grade, well differentiated, non-invasive SCC (Figure 15H), along with a dysplasia (Figure 15I) and, finally, high-grade, well differentiated, with moderate invasive characteristics SCC (Figure 15J). Interestingly, Group VI mice manifested acanthosis and dysplasia but no neoplastic transformation (Figure 15K).

### 3.5. Machine Learning: Hierarchical Clustering (HCL) and K-Means

HCL did not manifest any significant classification of the estimated variables. However, k-means classification manifested some interesting results. For the k-means analyses, two separate analyses were performed. The first included the k-means of the estimated experimental group-related variables, with respect to time (Figure 16A,B). This analysis included the calculation of variables clustered together with respect to time (Figure 16A), along with their centroids (Figure 16B). Similarly, the second included the k-means of the estimated experimental time-related variables, with respect to the experimental group measurements (Figure 16C,D). This analysis, respectively, included the calculation of variables clustered together, with respect to experimental groups (Figure 16C), along with their centroids (Figure 16D). The clusters are expected to show common patterns of variable behavior. In the case of group-related clustering (Figure 16A,B), we have found that clusters 1, 3, 5 and 8 manifested common patterns. In addition, in the case of time-related clustering, clusters 1, 4, 5, 6, will be discussed further on.

#### 3.5.1. The Analysis of K-Means Clusters with Respect to Time

The discovered clusters were analyzed for possible patterns in the evaluated factors. The clusters presented in Figure 16A–C concerned the examination of variables with respect to time. From the analysis, it appeared that certain patterns arose from the k-means clusters. Cluster 1 (Figure 16A–C) showed that glutathione in groups III and IV could be described by a logarithmic function (Figure 17A). This indicated that glutathione functions in a time-dependent manner, reaching a threshold after five months of treatment. Yet, body weight (BW) for groups I, II, IV, V manifested a linear increase (Figure 17B), which confirmed that the diabetic mice (groups III and VI) were not gaining weight. This was confirmed by Cluster 3, where the algorithm successfully classified groups III and VI together (Figure 17C). Interestingly, the k-means algorithm classified together groups IV and V, with respect to their hydration levels (Figure 17D). Finally, another interesting result came from the classification of TEWL measurements, where it appeared that month 3 was a critical turning point for groups IV and V (Figure 17E). Interestingly, it appeared that TEWL, for the diabetes Type 2 mice reached a maximum on month 3, surpassed that threshold, and then continued rising.

#### 3.5.2. The Analysis of K-Means Clusters, with Respect to the Experimental Groups

On the other hand, clustering, with respect to experimental groups, was expected to reveal clusters that would manifest a pattern for groups I–III and IV–VI. This part of our analysis included the evaluation of possible patterns in clusters presented in Figure 18A–F, which concerned the examination of variables with respect to the experimental groups. It was found that oxidative stress manifested a descending pattern in groups IV and VI, indicating a protective role of diabetes against UV ROS creation (Figure 18A). Furthermore, as expected, glucose and melanin manifested symmetrical pattern with respect to groups (Figure 18B). Interestingly, sebum manifested lower levels in irradiated mice, irrespective of their diabetic status, and, in particular, were found to be significantly different, as compared to non-irradiated mice (Figure 18C). Finally, glucose and hydration were clustered together and manifested similar results in the first month of the experiment, indicating that our experimental model started from a common reference (Figure 18D). At the same time, TEWL was divided into two main groups, with similar levels within the two groups. In particular, groups IV–VI manifested significantly higher levels of TEWL, as compared to groups I–III (Figure 18E). It was particularly noteworthy that the pattern revealed for TEWL levels was exactly symmetrical with that manifested by sebum. Finally, weight loss was observed in all groups (irrespectively of irradiation), indicating that diabetes was the decisive factor in weight loss (Figure 18F).

### 3.6. Regressions

#### 3.6.1. Oxidative Stress (OS) and TEWL

The next step in our analysis was the identification of correlations between variables at specific time points, and with respect to experimental groups. Our first observation concerned the significant correlations between oxidative stress on the fifth month (OS) and TEWL (Figure 19). Significant correlations were observed between OS on the fifth month (OS_5) and TEWL on month 3 (TEWL_3) (Figure 19A), TEWL on month 4 (TEWL_4) (Figure 19B), TEWL on month 5 (TEWL_5) (Figure 19C) and TEWL on month 6 (TEWL_6) (Figure 19D). In all cases the highest TEWL and OS levels were observed for groups IV and V, indicating that irradiation was the decisive factor in OS. Yet, Group VI, which corresponds to irradiated mice with diabetes, manifested the lowest levels, as compared to month 3 (Figure 19A), increased on month 4 (Figure 19B) and remained constant during months 5 (Figure 19C) and 6 (Figure 19D). It was noteworthy that diabetes consisted of a protective factor for the presence of OS in irradiated mice.

#### 3.6.2. Melanin and Skin Thickness

Following the previous observations, we have examined the significant correlations between skin thickness (Thickness) and melanin (Mel) in the fifth month (Mel_5) (Figure 20). Significant correlations were observed between melanin in the fifth month (Mel_5) and skin thickness in month 1 (Thickness_1) (Figure 20A), skin thickness in month 2 (Thickness_2) (Figure 20B) and skin thickness in month 3 (Thickness_3) (Figure 20C). In all cases, two separate groups were formed; on one hand, groups I, II, III formed a cluster manifesting lower levels of melanin and skin thickness, while groups IV, V, VI formed a second cluster manifesting higher levels of melanin and skin thickness. Thus, from these observations, we could hypothesize that skin thickness and melanin manifested different patterns with respect to irradiation, since the main difference between the two formed clusters was the presence or non-presence of UV. Furthermore, it appeared that skin thickness and melanin are two variables that are proportional to each other, as higher skin thickness levels indicate higher melanin levels.

#### 3.6.3. Oxidative Stress (OS) and Skin Thickness

Similar to the correlations found between TEWL and OS, we have also identified significant correlations between skin thickness and OS. The connection was not apparent, meaning that it is not clear how skin thickness can be associated with the appearance of oxidative stress. Yet, we have examined the significant correlations between skin thickness (Thickness) and oxidative stress (OS) in the fifth month (OS_5) (Figure 21). Significant correlations were observed between OS in the fifth month (OS_5) and skin thickness in month 3 (Thickness_3) (Figure 21A), skin thickness in month 4 (Thickness_4) (Figure 21B), skin thickness in month 5 (Thickness_5) (Figure 21C) and skin thickness in month 6 (Thickness_6) (Figure 21D). In all cases, the highest levels of skin thickness and OS were manifested by groups IV and V, while the lowest levels were manifested by groups I, II and III. Interestingly, Group VI, remained in-between groups I, II, III and IV, V, with a decreasing tendency from month 3 to month 6. This pattern has been marked in Figure 21, where Group VI* appears to decrease in value as we move from one month to the next. As in the case of TEWL association with OS, Type 1 diabetes manifested a protective effect, with respect to OS.

#### 3.6.4. Glutathione (Glut) and Skin Hydration (Hydr) 

Glutathione levels manifested significant correlations with skin hydration levels and keratinocyte mass. Glutathione in month 3 manifested significant correlations with skin hydration in month one (Figure 22A) and keratinocyte mass in month three (Figure 22B). In addition, glutathione in month five manifested significant correlation to keratinocyte mass in month five (Figure 22C). The high correlation levels were, on the one hand, expected due to the points (four) used for regression. On the other hand, interestingly, glutathione did not manifest any significant correlations with any other estimated variables.

##### 3.6.5. 3D Regressions: Glutathione 

Besides the correlations, we have also investigated the presence of regression among three variables. Our analysis has revealed significant correlations between the estimated variables. Significant correlations were observed between skin thickness (Thickness) of Group I, sebum (SE) for Group II, glutathione (Glut) for Group IV (Figure 23A) and glutathione for Group VI (Figure 23B). Significant correlations were observed between skin thickness (Thickness) of Group I, sebum (SE) for Group III, glutathione (Glut) for Group IV (Figure 23C) and glutathione for Group VI (Figure 23D). Furthermore, significant correlations were observed between skin thickness (Thickness) of group III, sebum (SE) for Group III, glutathione (Glut) for Group IV (Figure 23E) and glutathione for Group VI (Figure 23F). In addition, significant correlations were observed between skin thickness (Thickness) of group III, sebum (SE) for Group V, glutathione (Glut) for Group IV (Figure 23G) and glutathione for Group VI (Figure 23H). Finally, significant correlations were observed between skin thickness (Thickness) of Group III, sebum (SE) for Group VI, glutathione (Glut) for Group IV (Figure 23I) and glutathione for Group VI (Figure 23J). All data were modelled with a second order polynomial function, and yet the most interesting observation was that the function formed a concave upwards graph for thickness, sebum, and glutathione (Figure 23A,B,G,H). The function turned the concave downwards, for thickness, sebum, and glutathione, in other cases (Figure 23C–F,I,J).

### 3.7. ROC Analysis 

Our final step in the present analysis was the investigation of potential factors that could participate, with respect to the presence of tumors. Due to the nature of our experimental model, tumor presence was present in one group (Group IV). Therefore, it was imperative to examine the effect of multiple estimated variables simultaneously, with respect to time, or multiple experimental groups, with respect to time (i.e., the months of the experiment). The investigation of single months or single groupings, with respect to all variables, did not manifest any significant results. Therefore, we performed a multi-class analysis of our variables. Interestingly, significant results were manifested when examining multiple months, with respect to tumor presence. In particular, months 5 and 6 (Figure 24A), 0, 1 and 6 (Figure 24B), 0, 2 and 6 (Figure 24C), 0, 5 and 6 (Figure 24D), 1, 4 and 6 (Figure 24E), 1, 5 and 6 (Figure 24F), 2, 5 and 6 (Figure 24G), 3, 5 and 6 (Figure 24H) and 4, 5 and 6 (Figure 24I) manifested significant separation between animals that grew a tumor, as compared to those who did not.

## 4. Discussion

The relative risks associated with the development of most cancers in patients with diabetes range from a 1.2- to 2-fold increase [30]. To date, only prostate cancer has been identified as having a lower risk in diabetic patients. A retrospective cohort study conducted in Taiwan indicated a reduced risk of developing melanoma in the diabetic cohort, although this finding was not statistically significant [31]. Furthermore, the incidence rate and risk of developing non-melanoma skin cancer were found to be significantly higher in older adults with diabetes [32].

An animal experimental model was developed to investigate the effects of Type 1 and Type 2 Diabetes Mellitus on the development of skin cancer under UV irradiation treatment. High doses of streptozotocin induced Type 1 diabetes in the mouse model used. The manifestation of Type 1 diabetic skin occurred when glucose levels reached a maximum of 500 mg/dL, resulting in polydipsia, polyphagia, and polyuria. Intriguingly, it was discovered that mice with induced Type 1 Diabetes Mellitus were protected from developing any form of neoplasm following exposure to UV radiation, in contrast to all other groups, which did experience some form of skin transformation, including tumors. The oxidative stress (OS) in the stratum corneum was observed to decrease after five months, with the irradiated skin displaying precancerous lesions that developed into carcinomas by the eighth month of the experiment (Figure 14 and Figure 15). Conversely, irradiated Type 1 diabetic skin did not exhibit papillomata, hyperkeratosis, or carcinomas (Figure 12, Figure 14 and Figure 15). To the best of our knowledge, this is the inaugural report of such a study, indicating a lack of prior knowledge on the topic. Glucose levels are linked to the skin condition in diabetic patients, such as an incomplete skin barrier, dry skin, thinning, and aging [32]. Diabetic skin possesses distinctive characteristics that differentiate it from normal skin, including significantly reduced trans-epidermal water loss and a pronounced dry morphology. Furthermore, reductions in skin thickness and elasticity were observed, which are characteristic signs of prematurely aged skin (Figure 3, Figure 4, Figure 5 and Figure 8C). Consequently, in the stratum corneum, the hydrophilic protective antioxidant molecule glutathione was found to be reduced, and uric acid levels were found to be elevated (Figure 9 and Figure 10).

### 4.1. Oxidative Stress and Glutathione

Irradiation (specifically UV radiation in this study) and the resultant oxidative stress are recognized as significant contributors to the development of tumors, particularly skin cancer [33]. Conversely, glutathione is known for its protective role against cellular stress caused by reactive oxygen species [34]. In this study, an intriguing pattern was observed with both glutathione and oxidative stress. Specifically, glutathione levels were lower in the experimental group and notably in Group VI, wherethe Type 1 diabetic mice displayed the lowest glutathione levels at month 6 (Figure 10). Similarly, oxidative stress followed a parallel trend, although the differences were not statistically significant (Figure 11).

The oxidative stress levels in the stratum corneum did not indicate elevated oxidative stress when compared to the control levels, suggesting no significant difference. Furthermore, skin from Type 1 diabetic mice, when exposed to chronic UV radiation, was observed to be more dehydrated than normal irradiated skin, also showing reduced trans-epidermal water loss (Figure 3 and Figure 4).

Tissue oxidative stress has been extensively researched in the context of skin cancer, providing valuable insights into the mechanisms driving skin cancer development [35,36,37,38]. However, there remains a gap in understanding the specific roles of oxidative stress (OS) and redox status in skin cancer. It is established that exposure to UV radiation triggers the production of H_2_O_2_ by keratinocytes, which in turn leads to the consumption of catalase [39], a process also associated with an increase in glutathione activity [40]. Interestingly, it has been observed that diabetes appears to reverse the effects of oxidative stress and alter glutathione levels. Specifically, mice with Type 1 diabetes subjected to irradiation exhibited the lowest levels of glutathione, along with reduced keratinocyte numbers and oxidative stress levels, compared to their non-diabetic and Type 2 diabetic counterparts (Figure 10 and Figure 11). To our knowledge, this phenomenon has not been previously documented. This observation may not be fully explained in isolation but suggests that a combination of factors could potentially elucidate the presence or absence of tumors. Recent research has proposed that the interplay of catalase activity, keratinocyte-induced oxidative stress by UVB, and H_2_O_2_ production, could be key in understanding tumor induction on the skin [37]. This hypothesis aligns with our findings, indicating that an assessment of these combined parameters can differentiate between mice that develop tumors and those that do not.

Although UV-induced oxidative stress, particularly lipid peroxidation of cell membranes, is known to contribute to skin transformation, and diabetes negatively impacts epithelial cells [39,40,41,42], our study found that Type 1 diabetes in mice resulted in a decrease in keratinocytes, as well as in levels of glutathione and oxidative stress. Remarkably, the lowest levels of glutathione were observed in the irradiated Type 1 diabetic mice, suggesting that diabetes alone can enhance glutathione levels, yet this effect is inverted in the presence of UV radiation. Our findings confirm that Type 1 diabetic mice were protected from oxidative stress and glutathione depletion, compared to their irradiated non-diabetic counterparts.

FT-IR analysis of the skins of mice exposed to ultraviolet radiation—comprising non-diabetic, Type 2 diabetic, and Type 1 diabetic subjects—revealed that the environment of the keratinocytes in the epidermis of non-diabetic and Type 2 diabetic mice became more lipophilic, compared to that of Type 1 diabetic mice, whose keratinocyte environment resembled that of normal, non-irradiated mice. Furthermore, oxidative stress, as indicated by the spectrum of aldehydes associated with lipid peroxidation, was found to be increased in the case of ultraviolet-irradiated non-diabetic and Type 2 diabetic mice, in contrast to those irradiated with Type 1 diabetes, where oxidative stress levels were observed to be within normal ranges [43].

### 4.2. Keratinocytes

A notable observation in our study was the initial occurrence of hyperkeratosis due to UV radiation exposure, which subsequently diminished as the skin of diabetic subjects thinned (Figure 5). Additionally, there was a significant reduction in sebum lipids and skin hyperpigmentation (Figure 8A,B). It is established that healthy skin, following prolonged exposure to UV radiation, typically undergoes thickening, experiences reduced hydration, exhibits increased levels of trans-epidermal water loss, and shows a rise in the concentration of sebaceous lipids and melanin [44,45]. These findings are consistent with our observations, where we recorded similar skin morphology changes in our experimental model (Figure 3, Figure 4, Figure 5 and Figure 8A,B).

### 4.3. Type 2 and 1 Diabetic Skin

The data from the study indicate that Type 2 diabetes induced by streptozotocin (20 mg/kg) mirrors the characteristics of normal mice across almost all measured parameters, both without (Figure 3, Figure 4, Figure 5, Figure 6, Figure 7, Figure 8C, Figure 11, Figure 13 and Figure 15) and with exposure to ultraviolet light (Figure 3, Figure 4, Figure 5, Figure 6, Figure 7, Figure 8C, Figure 11, Figure 12, Figure 14 and Figure 15). The only observed difference pertains to the onset time in relation to the size of the papillomata, the volume of cSCC, or the intensity of symptoms observed in histopathological evaluations (Figure 12, Figure 14 and Figure 15, respectively). As previously mentioned, and in the evaluation of skin using FT-IR spectra following exposure to ultraviolet light, analogous data were observed [43].

Conversely, different data were obtained from Type 1 diabetes, compared to the other two mice groups. The trans-epidermal water loss (Figure 3), hydration (Figure 4), skin thickness (Figure 5), mice weight (Figure 6), sebum (Figure 8A), keratinocyte acquisition via stripping (Figure 8C), levels of uric acid and glutathione (Figure 9 and Figure 10, respectively), clinical appearance (Figure 12, Figure 13 and Figure 14), and histopathological evaluation (Figure 15) were significantly different in Type 1 diabetes, compared to normal skin and Type 2 diabetes in the majority of cases, both with and without the application of ultraviolet light. This also applied to the acquisition of FT-IR spectra following the application of ultraviolet light [43].

It is established that endogenous aging of skin cells acts as an inhibitory factor in tumorigenesis [46]. The fact that Type 1 diabetes causes significant endogenous aging of the skin [30] leads us to question if this significantly contributes to the skin’s ability to prevent carcinogenesis.

It should be highlighted that the anti-diabetic medication, Metformin, has been linked to a decreased risk of skin cancer [31]. Type 1 diabetes and associated hyperglycemia, as well as skin carcinogenesis, are all associated with the presence of oxidative stress [47]. Although the mechanisms of action of Metformin are complex and not entirely understood, the Type 1 diabetes/skin carcinogenesis animal model offers a valuable tool for exploring the mechanisms of skin cancer suppression by diabetes and the inhibition of diabetes by Metformin.

## 5. Conclusions

In this study, the impact of UV irradiation on diabetic skin was explored using an animal model. Changes in hydrophilic antioxidants within the stratum corneum following UV radiation exposure were discovered, and notably a significant reduction in glutathione levels, and significant increase in uric acid and oxidative stress. A particularly notable finding was that, after five months, the irradiated normal and Type 2 diabetic skin developed papillomata and precancerous lesions, and that, by the sixth month, squamous cell carcinoma was evident. In contrast, mice subjected to a high dose of streptozotocin to induce Type 1 diabetes did not develop any papillomata and cancerous lesions, indicating a potential protective role of diabetes against the neoplastic transformation of the skin.

## Figures and Tables

**Figure 1 cancers-16-01507-f001:**
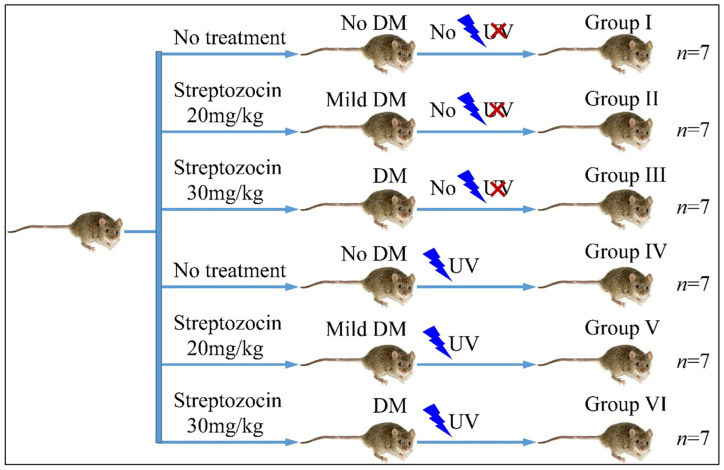
Diagrammatical representation of the experimental design (Legend: DM: Diabetes Mellitus, UV: Ultra-Violet).

**Figure 2 cancers-16-01507-f002:**
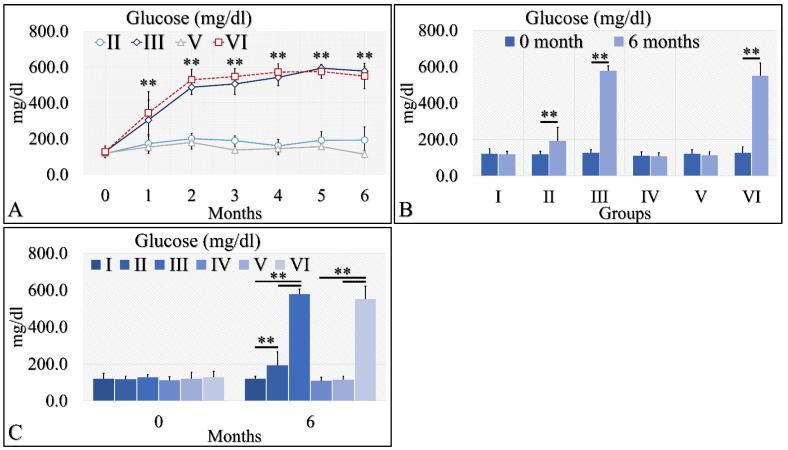
Diabetes induction was successful in the II, III and VI mice groups (II, III, V, VI), where glucose levels were significantly higher, as compared to the month of reference (0 month) (**A**). Significant differences were observed between month 0 and month 6 in groups II, III and VI (**B**). In addition, groups II and III manifested significantly higher glucose levels, as compared to the control experiment at 6 months, and group VI manifested significantly higher glucose levels, as compared to the irradiated reference (Group IV) and irradiated Type 2 diabetic mice (Group V) (C) (Legend: I: No treatment, no diabetes induction, no irradiation, II: Type 2 diabetes induction, no irradiation, III: Type 1 diabetes induction, no irradiation, IV: No diabetes induction, irradiation, V: Type 2 diabetes induction, irradiation, VI: Type 1 diabetes induction, irradiation. In subfigure (**A**), ** indicates a significance of *p* << 0.01, as compared to the month 0. In subfigures (**B**,**C**), ** depicts a significance of *p* << 0.01).

**Figure 3 cancers-16-01507-f003:**
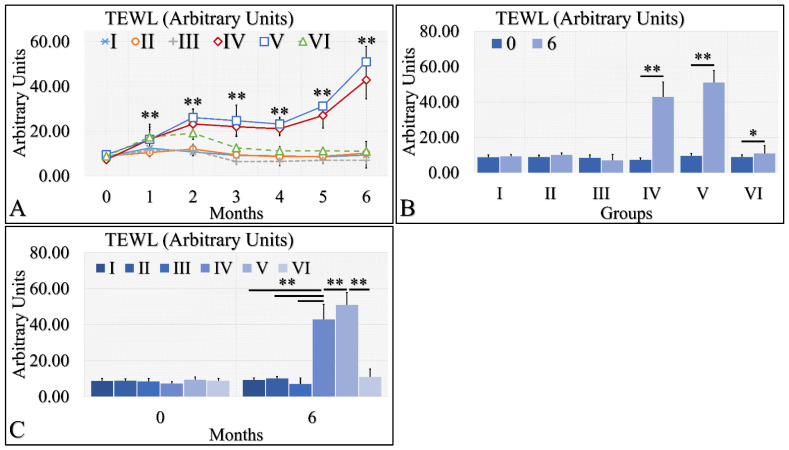
TEWL was observed mostly in groups IV and V. In particular, TEWL manifested an ascending pattern, with respect to time for groups IV and V (**A**). This result was also manifested in the comparison between months 0 and 6, where irradiated mice and irradiated mice with Type 2 diabetes manifested higher TEWL levels (**B**). Similarly, groups IV and V manifested significantly higher TEWL levels, as compared to all other groups (**C**) (Legend: I: No treatment, no diabetes induction, no irradiation, II: Type 2 diabetes induction, no irradiation, III: Type 1 diabetes induction, no irradiation, IV: No diabetes induction, irradiation, V: Type 2 diabetes induction, irradiation, VI: Type 1 diabetes induction, irradiation. In subfigure (**A**), ** indicates a significance of *p* << 0.01, as compared to the month 0 and to the other groups in the same month. In subfigures (**B**,**C**), ** depicts a significance of *p* << 0.01, * Depicts the significance of *p* < 0.05).

**Figure 4 cancers-16-01507-f004:**
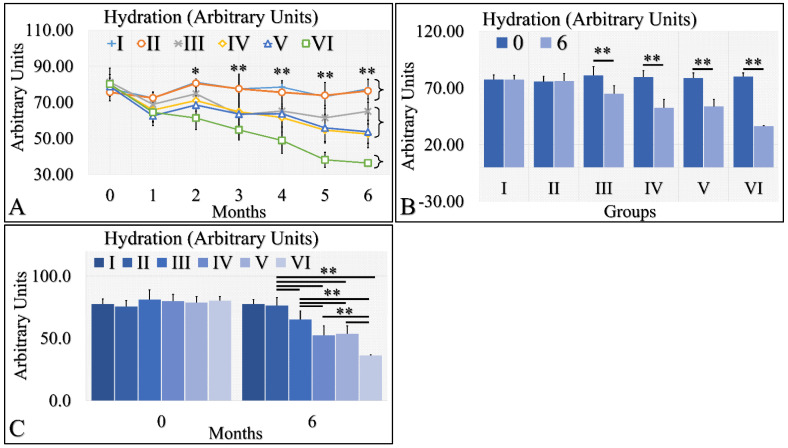
Hydration levels manifested three clusters; groups I and II, groups III, IV and V, and group VI manifested a significant decrease in hydration levels (**A**). Groups I and II did not manifest significant differences between month 0 and month 6, while groups III, IV, V and VI manifested significant decreases from month 0 to month 6 (**B**). Groups I and II manifested similar hydration levels at month 6, as well as groups IV and V (**C**). Groups III and VI manifested significantly different levels of hydration, as compared to all other (**C**). Hydration levels manifested a clear descending trend from the control group to the irradiated diabetic group (**C**) (Legend: I: No treatment, no diabetes induction, no irradiation, II: Type 2 diabetes induction, no irradiation, III: Type 1 diabetes induction, no irradiation, IV: No diabetes induction, irradiation, V: Type 2 diabetes induction, irradiation, VI: Type 1 diabetes induction, irradiation. In subfigure (**A**), ** indicates a significance of *p* << 0.01, as compared to the month 0 and to the other groups in the same month; and * indicates a significance of *p* < 0.05. In subfigures (**B**,**C**), ** depicts a significance of *p* << 0.01).

**Figure 5 cancers-16-01507-f005:**
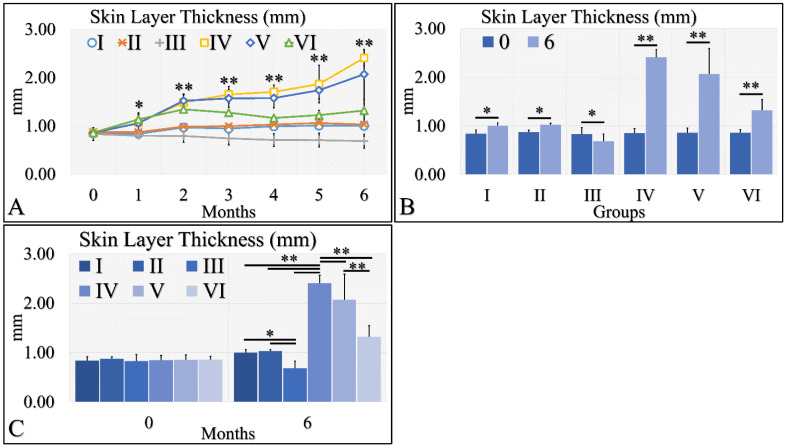
Skin thickness was evaluated and groups IV and V, manifested the higher skin thickness, as compared to all other groups (**A**). In addition, groups IV and V manifested an ascending trend from month 0 to month 6 (**A**). Furthermore, all groups, except Group III, presented significantly higher skin thickness between months 0 and 6 (**B**). Finally, when comparing all groups at month 6, significant differences were manifested between all groups, with the exception of groups I and II (**C**) (Legend I: No treatment, no diabetes induction, no irradiation, II: Type 2 diabetes induction, no irradiation, III: Type 1 diabetes induction, no irradiation, IV: No diabetes induction, irradiation, V: Type 2 diabetes induction, irradiation, VI: Type 1 diabetes induction, irradiation. In subfigure (**A**), ** indicates a significance of *p* << 0.01, as compared to the month 0 and the other groups in the same month; and * indicates a significance of *p* < 0.05. In subfigures (**B**,**C**), ** depicts a significance of *p* << 0.01, * Depicts the significance of *p* < 0.05).

**Figure 6 cancers-16-01507-f006:**
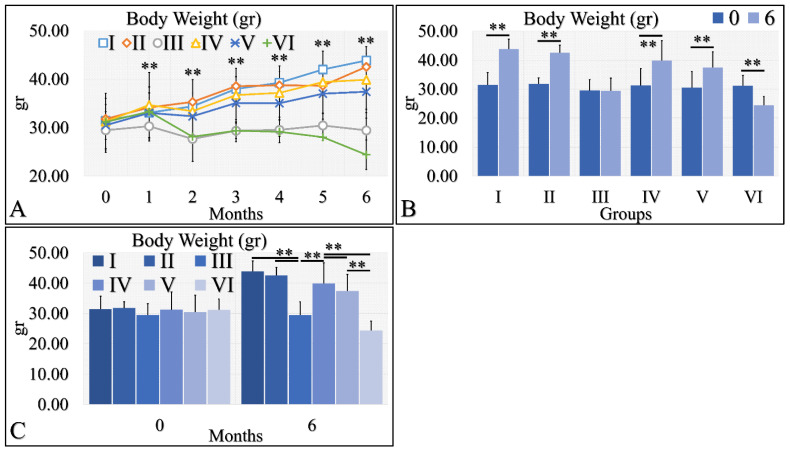
Body weight was evaluated and measured in all experimental groups. The irradiated Type 1 diabetic mice (Group VI) manifested the lowest body weight, as compared to all other groups (**A**). This was followed by the Type 1 diabetic mice (Group III), while all other groups manifested an increase in body weight, with respect to time (**A**). The comparison of month 0 and month 6 for all experimental cases showed that groups I and II manifested a significant increase in body weight, while Group III did not manifest any differences (**B**). Groups IV and V manifested a significant increase in body weight and, on the contrary, the irradiated Type 1 diabetic mice lost weight after 6 months of treatment (**B**). Comparing mice body weight at month 6, groups III and VI manifested significantly lower body weight, as compared to the other groups, while control mice manifested the higher body weight (**C**) (Legend: I: No treatment, no diabetes induction, no irradiation, II: Type 2 diabetes induction, no irradiation, III: Type 1 diabetes induction, no irradiation, IV: No diabetes induction, irradiation, V: Type 2 diabetes induction, irradiation, VI: Type 1 diabetes induction, irradiation. In subfigure (**A**), ** indicates a significance of *p* << 0.01, as compared to the month 0 and to the other groups in the same month; In subfigures (**B**,**C**), ** depicts a significance of *p* << 0.01).

**Figure 7 cancers-16-01507-f007:**
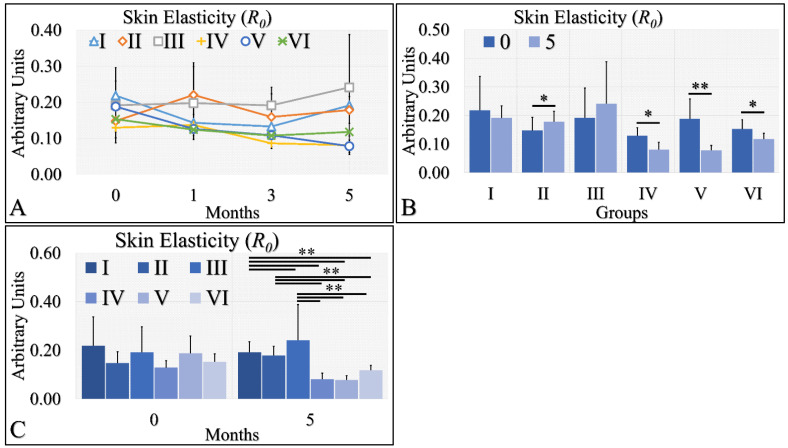
Skin elasticity measurements can be divided into two classes; the highest levels were manifested by groups I, II, III; however, there no significant difference among them (**A**). On the other hand, the lowest levels of skin elasticity were observed by groups IV, V and VI also, with no significant difference among them (**A**). Comparing the skin elasticity between months 0 and 5, all groups manifested significant differences, except for groups I and III (**B**). Comparing the skin elasticity values at month 5, it was confirmed that no significant differences appeared between groups I, II, III, as well as IV, V and VI (**C**). Yet, groups I, II and III manifested significant differences with respect to groups IV, V and VI (**C**) (Legend: I: No treatment, no diabetes induction, no irradiation, II: Type 2 diabetes induction, no irradiation, III: Type 1 diabetes induction, no irradiation, IV: No diabetes induction, irradiation, V: Type 2 diabetes induction, irradiation, VI: Type 1 diabetes induction, irradiation. * Depicts the significance of *p* < 0.05. ** depicts a significance of *p* << 0.01).

**Figure 8 cancers-16-01507-f008:**
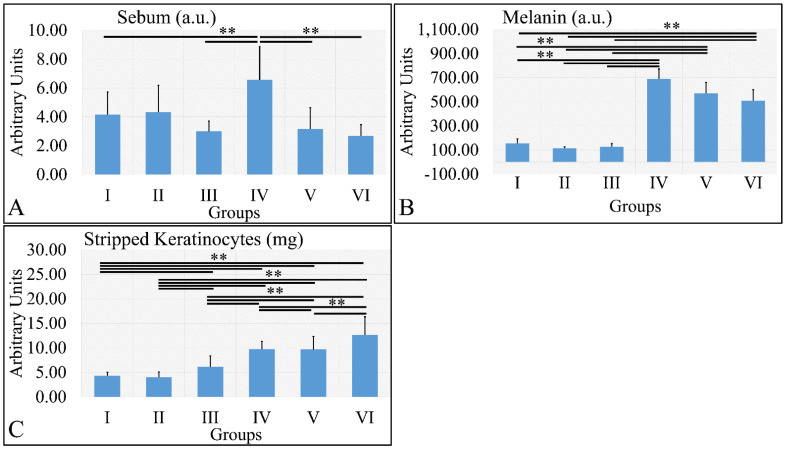
Sebum manifested significantly higher levels for Group IV, as compared to all other groups (**A**). Melanin levels were significantly higher in all irradiated groups, as compared to the non-irradiated groups (**B**). Stripped keratinocytes were significantly more abundant between all groups, as compared to each other, with the exception of groups I and II, where no significant differences were observed (**C**) (Legend: I: No treatment, no diabetes induction, no irradiation, II: Type 2 diabetes induction, no irradiation, III: Type 1 diabetes induction, no irradiation, IV: No diabetes induction, irradiation, V: Type 2 diabetes induction, irradiation, VI: Type 1 diabetes induction, irradiation. ** depicts a significance of *p* < 0.01).

**Figure 9 cancers-16-01507-f009:**
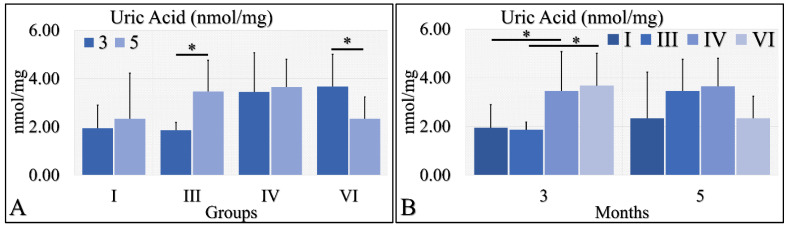
Uric acid measurements showed that Group III manifested significantly higher levels at month 5, as compared to month 3 (**A**), as well as Group VI manifested significantly lower levels at month 5, as compared to month 3 (**A**). No significant differences were observed at month 5 among all groups (**B**). Yet, irradiated control and irradiated diabetic mice manifested significantly higher levels of uric acid, as compared to non-irradiated groups (**B**) (Legend: I: No treatment, no diabetes induction, no irradiation, II: Type 2 diabetes induction, no irradiation, III: Diabetes induction, no irradiation, IV: No diabetes induction, irradiation, V: Type 2 diabetes induction, irradiation, VI: Diabetes induction, irradiation. * depicts a significance of *p* < 0.05).

**Figure 10 cancers-16-01507-f010:**
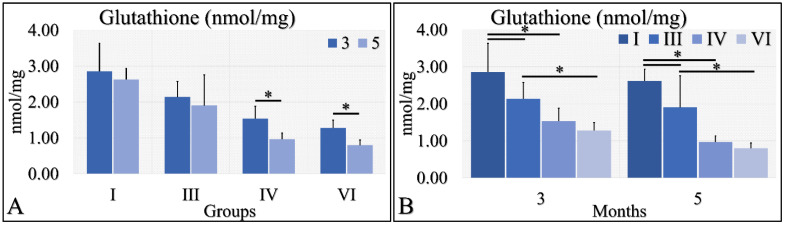
Glutathione measurements showed that Group IV manifested significantly higher levels at month 3, as compared to month 5 (**A**); and group VI manifested significantly higher levels at month 3, as compared to month 5 (**A**). When comparing the experimental groups amongst them, we have found that, in both months, control mice (Group I) manifested higher glutathione levels, as compared to diabetic (Group III) and irradiated diabetic (group VI) mice; we also found that diabetic (Group III) mice manifested significantly higher levels, as compared to irradiated diabetic (Group VI) mice (**B**) (Legend: I: No treatment, no diabetes induction, no irradiation, II: Type 2 diabetes induction, no irradiation, III: Diabetes induction, no irradiation, IV: No diabetes induction, irradiation, V: Type 2 diabetes induction, irradiation, VI: Diabetes induction, irradiation. * depicts the significance of *p* < 0.05).

**Figure 11 cancers-16-01507-f011:**
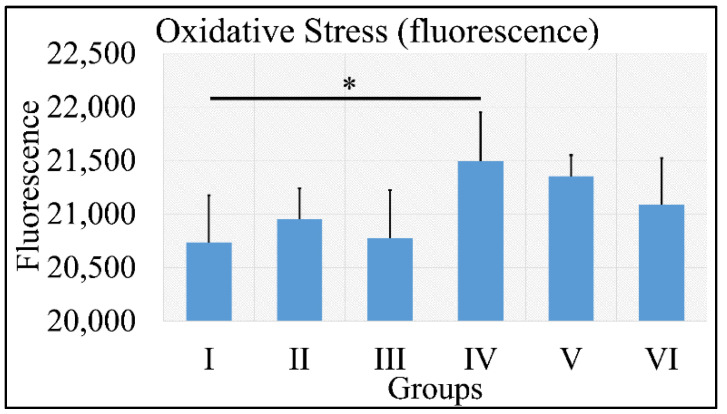
ROS measurements manifested that no significant differences were found among all groups, with the exception of Group IV, which manifested significantly higher ROS levels, as compared to Group I (Legend: I: No treatment, no diabetes induction, no irradiation, II: Type 2 diabetes induction, no irradiation, III: Type 1 diabetes induction, no irradiation, IV: No diabetes induction, irradiation, V: Type 2 diabetes induction, irradiation, VI: Type 1 diabetes induction, irradiation. * depicts the significance of *p* < 0.05).

**Figure 12 cancers-16-01507-f012:**
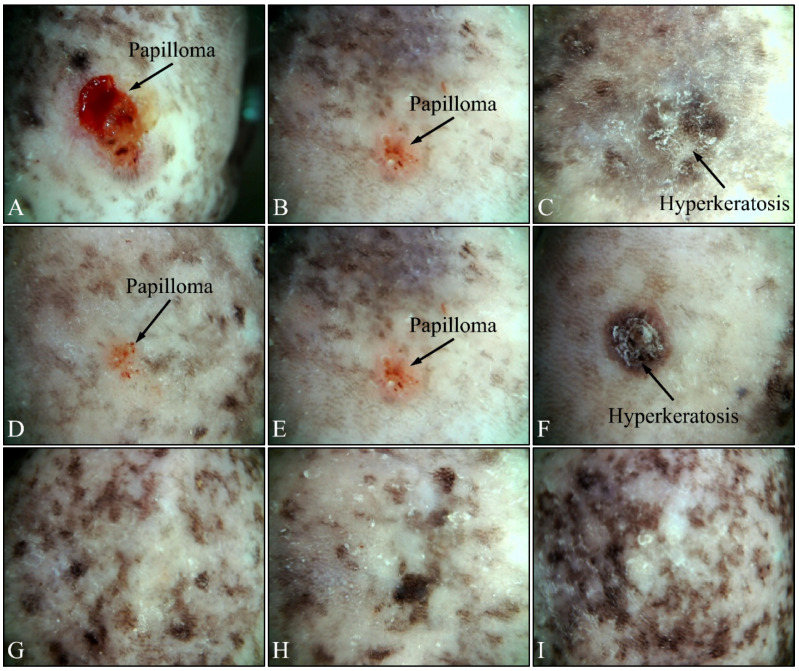
Dermatoscopic evaluation of mice experimental groups. Control irradiated mice manifested the presence of papilloma and hyperkeratosis after six months of treatment (**A**–**C**). In addition, irradiated Type 2 diabetic mice also manifested papilloma and hyperkeratosis, yet to a lower extent, as compared to the control group (**D**–**F**). On the contrary, irradiated Type 1 diabetic mice did not manifest any papilloma or hyperkeratosis (**G**–**I**).

**Figure 13 cancers-16-01507-f013:**
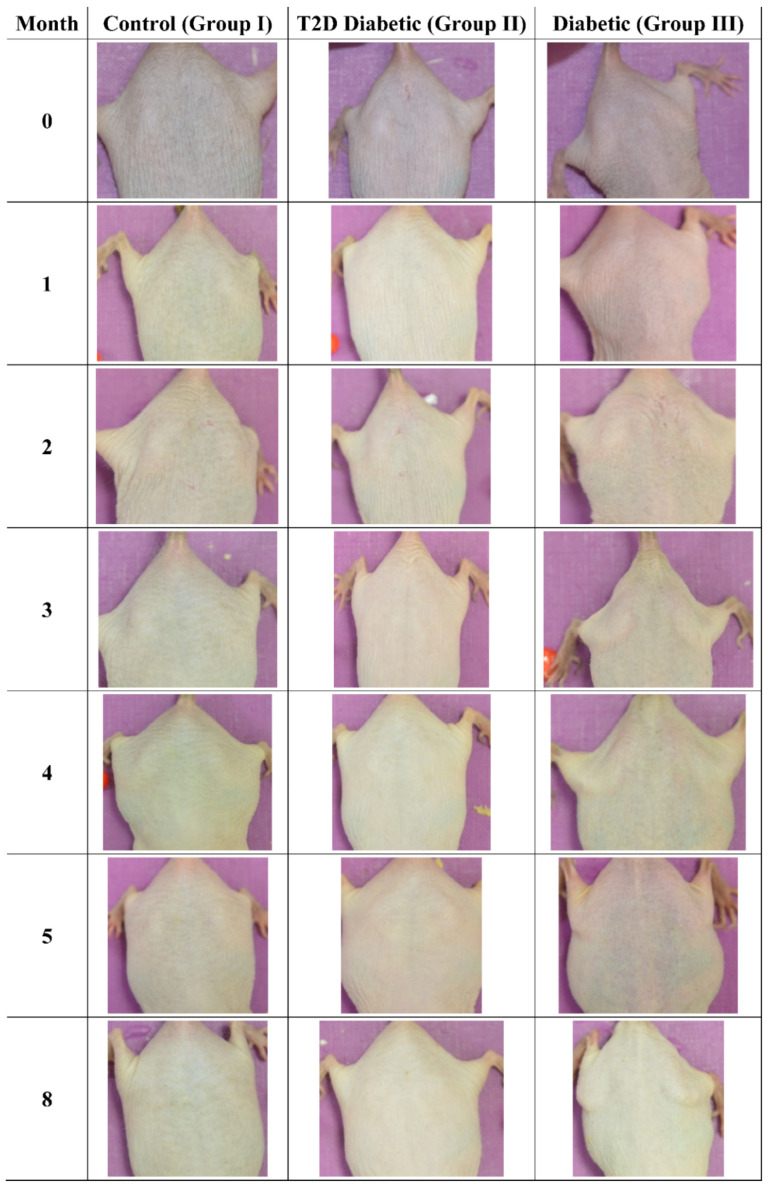
Photographic evaluation of UV irradiation effects on the experimental model. Non-irradiated mice manifested no skin benign or malignant transformations. (T2D: Type 2 Diabetes, Diabetic: Type 1 Diabetes).

**Figure 14 cancers-16-01507-f014:**
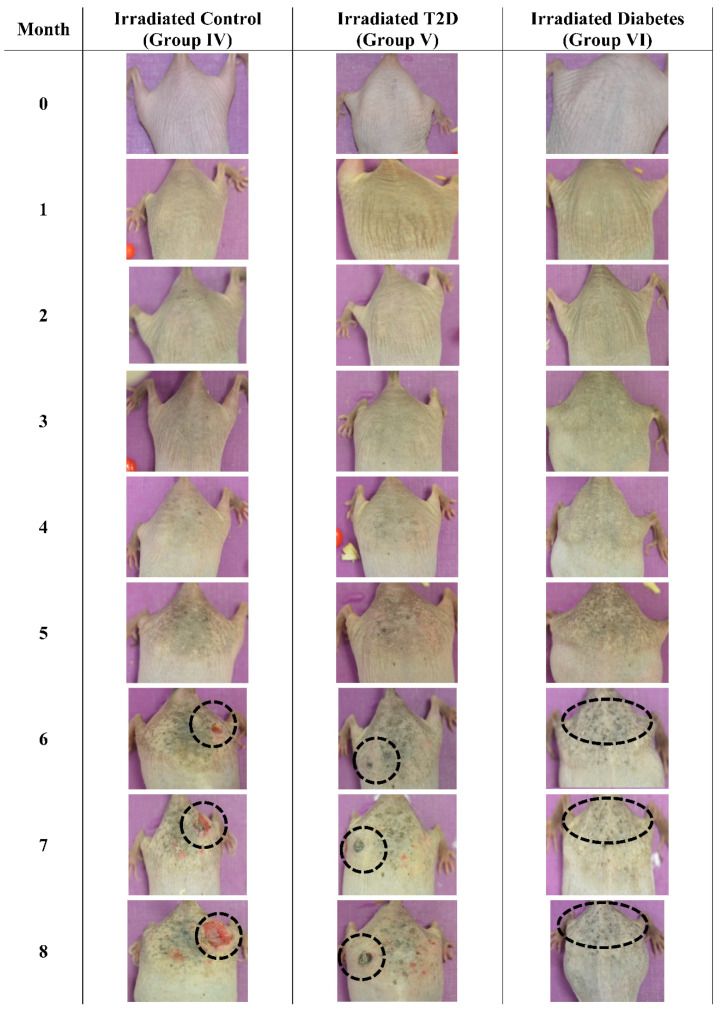
Photographic evaluation of UV irradiation effects on the experimental model. Irradiated control mice (no induced diabetes) manifested squamous cell carcinoma (dashed line), while irradiated Type 2 diabetic mice manifested milder squamous cell carcinoma (dashed line). On the other hand, irradiated Type 1 diabetic mice did not manifest any serious skin aberrations (dashed line) (T2D: Type 2 Diabetes, Diabetes: Type 1 diabetes).

**Figure 15 cancers-16-01507-f015:**
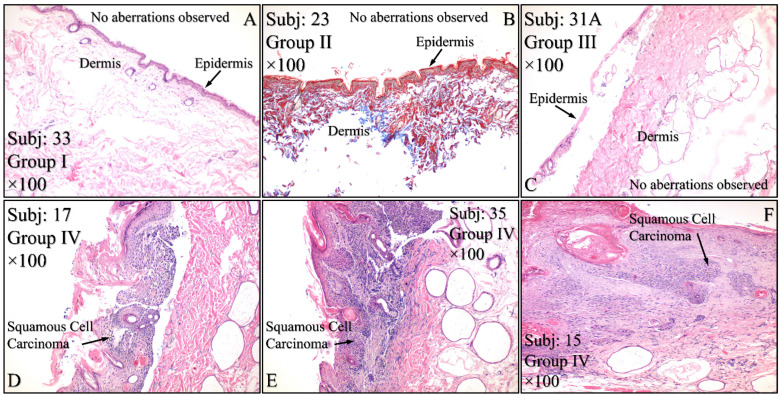
Histological evaluations of the experimental groups. Control animals (Group I) manifested no cytological transformations (**A**), as well as those animals in Group II (**B**) and Group III (**C**). On the other hand, diabetes-free irradiated mice (Group IV) manifested squamous cell carcinoma (SCC), which was manifested as a high-grade, well-differentiated, non-invasive SCC (**D**), high-grade, well-differentiated, invasive SCC (**E**) and high-grade, moderately differentiated, invasive SCC (**F**). Similar results were manifested by animals in Group V (irradiated with Type 2 diabetes), where mice were presented with invasive high-grade, well-differentiated SCC (**G**), dysplastic (**H**) and high-grade, well-differentiated, non-invasive SCC (**I**) and high-grade, well differentiated, moderatelly invasive SCC (**J**). Interestingly, Group VI mice (irradiated with Type 1 diabetes) manifested acanthosis and dysplasia but no neoplastic transformation (**K**) (Legend: I: No treatment, no diabetes induction, no irradiation, II: Type 2 diabetes induction, no irradiation, III: Type 1 diabetes induction, no irradiation, IV: No diabetes induction, irradiation, V: Type 2 diabetes induction, irradiation, VI: Type 1diabetes induction, irradiation, Subj: Subject).

**Figure 16 cancers-16-01507-f016:**
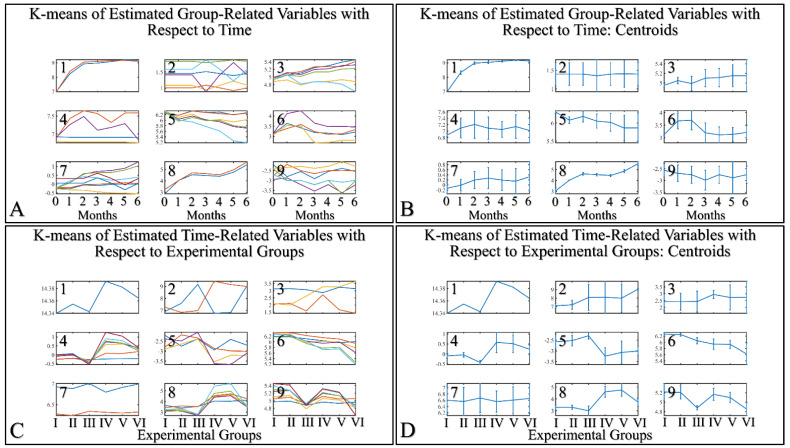
K–means clustering. Two separate analyses were performed. The first included the k-means of the estimated experimental group–related variables with respect to time (**A**,**B**). The second included the k–means of the estimated time related variable with respect to the experimental groups (**C**,**D**). Each analysis included the complete datasets, which is the actual presentation of all data and their respective clusters (**A**,**C**) and the centroids (**B**,**D**).

**Figure 17 cancers-16-01507-f017:**
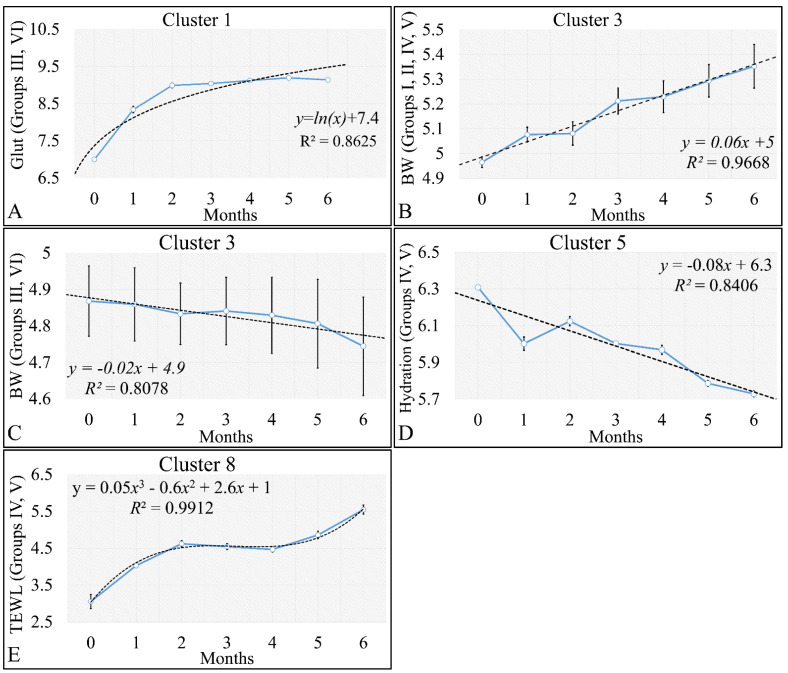
Analysis of the k–means clustering presented in Figure 16A–C. Certain patterns arose from the analysis of k–means clusters. Linear behavior was searched for all groups with respect to time. Cluster 1 showed that glutathione in groups III and IV (non–irradiated diabetic and irradiated non–diabetic mice) could be described by a logarithmic function (**A**). Yet, body weight (BW) for groups I, II, IV, V manifested a linear increase, which confirmed that the diabetic mice (groups III and VI) were not gaining weight (**B**). This was confirmed by cluster 3, where the algorithm successfully classified groups III and VI together (**C**). Interestingly, the k–means algorithm classified together groups IV and V with respect to their hydration levels (**D**). Finally, another interesting result came from the classification of TEWL measurements, where it appeared that month 3 was a critical point of turn for groups IV and V (**E**) (Legend: I: No treatment, no diabetes induction, no irradiation, II: Type 2 diabetes induction, no irradiation, III: Type 1 diabetes induction, no irradiation, IV: No diabetes induction, irradiation, V: Type 2 diabetes induction, irradiation, VI: Type 1 diabetes induction, irradiation).

**Figure 18 cancers-16-01507-f018:**
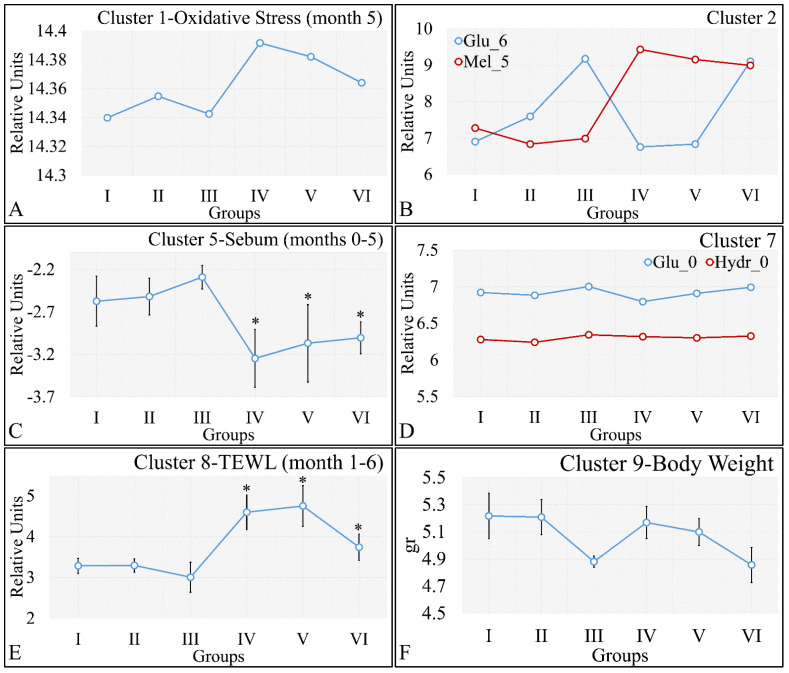
Analysis of the k-means clustering presented in Figure 18A–F. Certain patterns arose from the analysis of k-means clusters. Cluster 1 showed that oxidative stress manifested a descending pattern of groups IV–VI (**A**). Cluster 2 showed that glucose and melanin manifested symmetrical behavior for the experimental groups (**B**). At the same time, sebum in groups IV–VI manifested significantly higher levels, as compared to sebum in groups I–III (**C**). Hydration manifested similar levels for all groups, indicating that the experimental model started from the same reference level (**D**). Similarly, as in the case of sebum, groups IV–VI manifested significantly higher levels, as compared to groups I–III (**E**). Weight loss was affected by diabetes and not by irradiation, since groups I, IV, and II, V, as well as III and VI, manifested comparable levels (**F**) (Legend: I: No treatment, no diabetes induction, no irradiation, II: Type 2 diabetes induction, no irradiation, III: Type 1 diabetes induction, no irradiation, IV: No diabetes induction, irradiation, V: Type 2 diabetes induction, irradiation, VI: Type 1 diabetes induction, irradiation, TEWL: Trans-Epidermal Water Loss). * Depicts the significance of *p* < 0.05.

**Figure 19 cancers-16-01507-f019:**
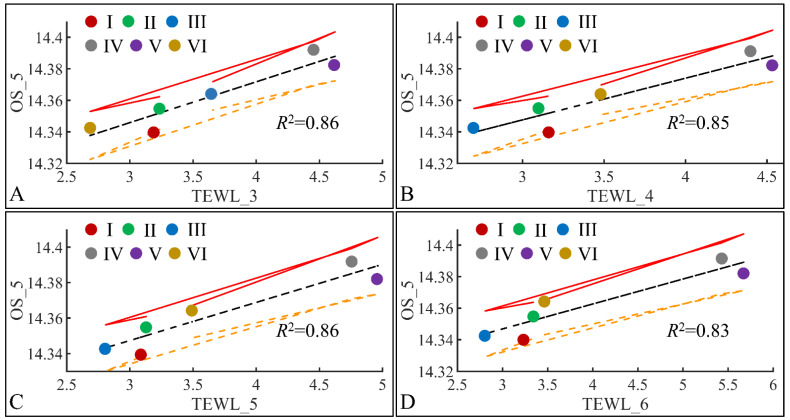
Regression analysis of evaluated variables, with respect to groups. The values of each variable, with respect to time, were regressed within experimental groups. Overall, oxidative stress (OS) on the fifth month (OS_5) was found to have significant correlations to TEWL for months 3 (TEWL_3) (**A**), 4 (TEWL_4) (**B**), 5 (TEWL_5) (**C**) and 6 (TEWL_6) (**D**) (Legend: I: No treatment, no diabetes induction, no irradiation, II: Type 2 diabetes induction, no irradiation, III: Type 1 diabetes induction, no irradiation, IV: No diabetes induction, irradiation, V: Type 2 diabetes induction, irradiation, VI: Type 1 diabetes induction, irradiation, TEWL: Trans-Epidermal Water Loss).

**Figure 20 cancers-16-01507-f020:**
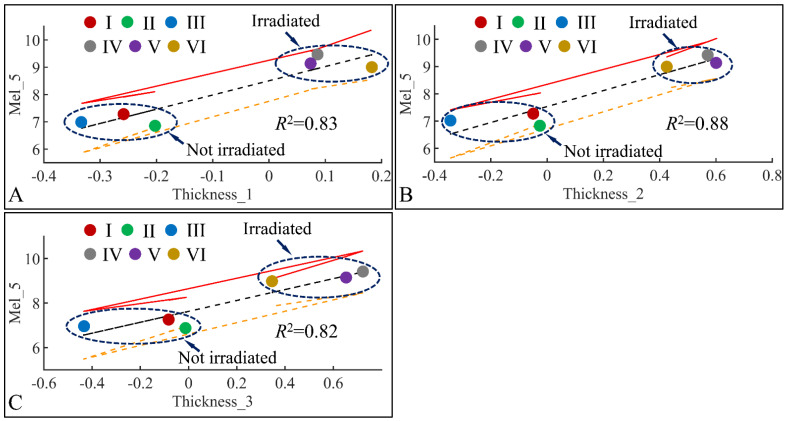
Regression analysis of evaluated variables with respect to groups. The values of each variable with respect to time was regressed within experimental groups. Overall, melanin (Mel) on the fifth month (Mel_5) was found to have significant correlations to skin thickness (Thickness) for months 1 (Thickness_1) (**A**), 2 (Thickness_2) (**B**) and 3 (Thickness_3) (**C**) (Legend: I: No treatment, no diabetes induction, no irradiation, II: Type 2 diabetes induction, no irradiation, III: Type 1 diabetes induction, no irradiation, IV: No diabetes induction, irradiation, V: Type 2 diabetes induction, irradiation, VI: Type 1 diabetes induction, irradiation).

**Figure 21 cancers-16-01507-f021:**
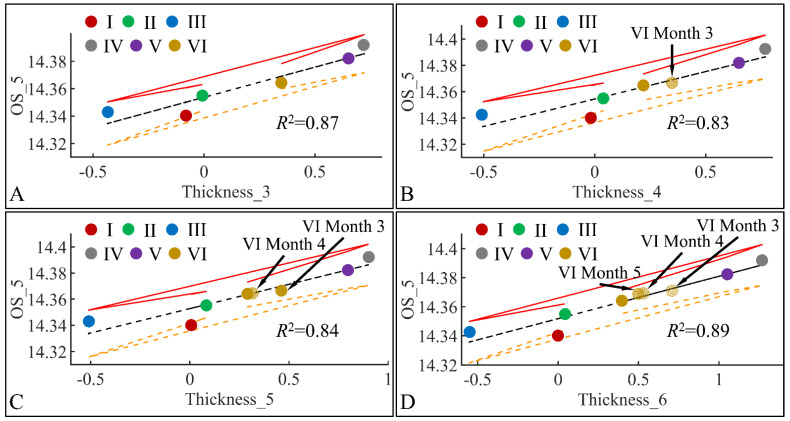
Regression analysis of evaluated variables, with respect to groups. The values of each variable, with respect to time, were regressed within experimental groups. Overall, oxidative stress (OS) on the fifth month (OS_5) was found to have significant correlations to skin thickness (Thickness) for months 3 (Thickness_3) (**A**), 4 (Thickness_4) (**B**), 5 (Thickness_5) (**C**) and 6 (Thickness_6) (**D**) (Legend: I: No treatment, no diabetes induction, no irradiation, II: Type 2 diabetes induction, no irradiation, III: Type 1 diabetes induction, no irradiation, IV: No diabetes induction, irradiation, V: Type 2 diabetes induction, irradiation, VI: Type 1 diabetes induction, irradiation).

**Figure 22 cancers-16-01507-f022:**
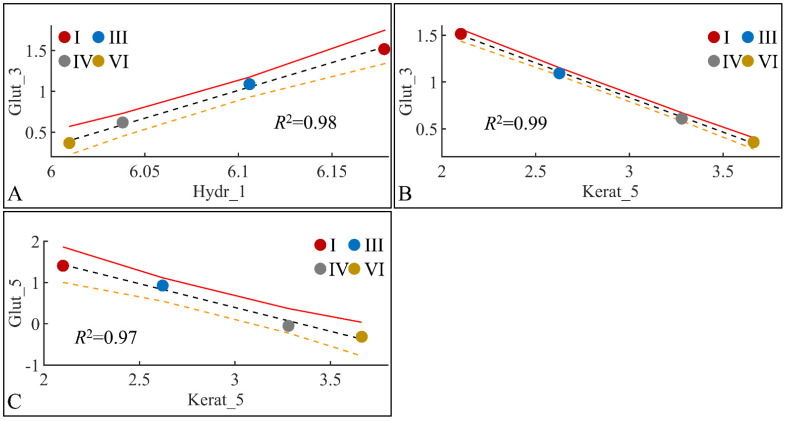
Regression analysis of evaluated variables, with respect to groups. The values of each variable, with respect to time, were regressed within experimental groups. Overall, glutathione (Glut) in the third month (Glut_3) was found to have significant correlations with skin hydration (Hydr) for months 1 (Hydr_1) (**A**), and keratinocyte mass (Kerat) for month 5 (Kerat_5) (**B**). At the same time, glutathione in month five (Glut_5) manifested significant correlation with keratinocyte mass in month five (Kerat_5) (**C**) (Legend: I: No treatment, no diabetes induction, no irradiation, II: Type 2 diabetes induction, no irradiation, III: Type 1 diabetes induction, no irradiation, IV: No diabetes induction, irradiation, V: Type 2 diabetes induction, irradiation, VI: Type 1 diabetes induction, irradiation).

**Figure 23 cancers-16-01507-f023:**
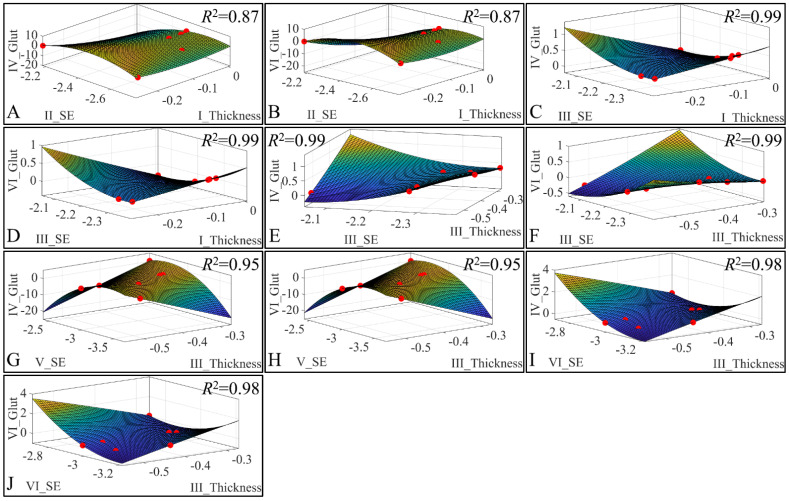
3D regression analysis of evaluated variables, with respect to groups. Significant correlations were observed between skin thickness (Thickness) of Group I, sebum (SE) for Group II, glutathione (Glut) for Group IV (**A**) and glutathione for Group VI (**B**). Significant correlations were observed between skin thickness (Thickness) of Group I, sebum (SE) for Group III, glutathione (Glut) for Group IV (**C**) and Glutathione for group VI (**D**). Further, significant correlations were observed between skin thickness (Thickness) of Group III, sebum (SE) for Group III, glutathione (Glut) for Group IV (**E**) and glutathione for Group VI (**F**). In addition, significant correlations were observed between skin thickness (Thickness) of Group III, sebum (SE) for Group V, glutathione (Glut) for Group IV (**G**) and glutathione for Group VI (**H**). Finally, significant correlations were observed between skin thickness (Thickness) of group III, sebum (SE) for Group VI, glutathione (Glut) for Group IV (**I**) and glutathione for Group VI (**J**) (Legend: I: No treatment, no diabetes induction, no irradiation, II: Type 2 diabetes induction, no irradiation, III: Type 1 diabetes induction, no irradiation, IV: No diabetes induction, irradiation, V: Type 2 diabetes induction, irradiation, VI: Type 1 diabetes induction, irradiation).

**Figure 24 cancers-16-01507-f024:**
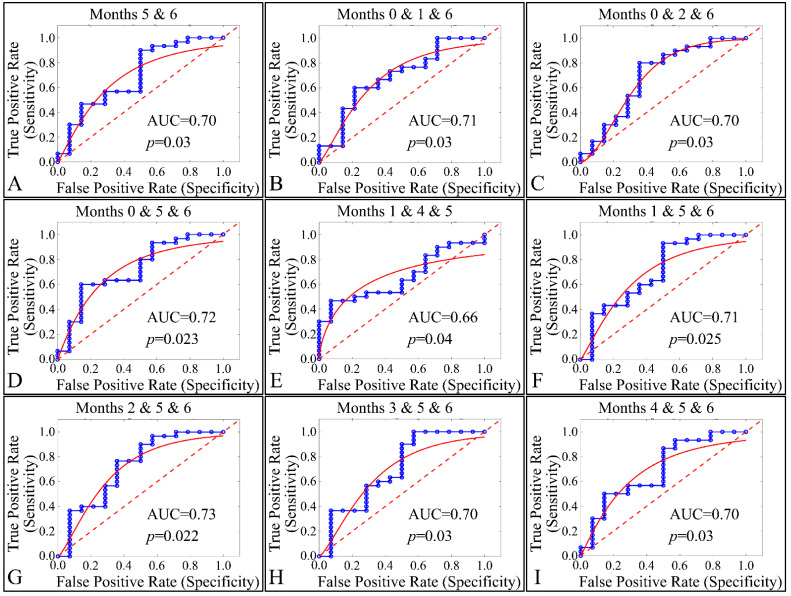
ROC analysis of estimated variables, with respect to time and the presence of tumor. Multi-class analysis was able to discriminate between tumor and non-tumor presenting animals, with respect to the duration of the experiment, i.e., months 5 and 6 (**A**), 0,1, 6 (**B**), 0, 2, 6 (**C**), 0, 5, 6 (**D**), 1, 4, 5 (**E**), 1, 5, 6 (**F**), 2, 5, 6 (**G**), 3, 5, 6 (**H**) and 4, 5, 6 (**I**) (Legend: I: No treatment, no diabetes induction, no irradiation, II: Type 2 diabetes induction, no irradiation, III: Type 1 diabetes induction, no irradiation, IV: No diabetes induction, irradiation, V: Type 2 diabetes induction, irradiation, VI: Type 1 diabetes induction, irradiation AUC: Area Under the Curve).

**Table 1 cancers-16-01507-t001:** The experimental groups involved in the study.

Group	Diabetes	Irradiation	Description
I	NO	NO	Control/Reference
II	T2D	NO	Type 2 diabetes induction (20 mg/kg streptozocin, low dose)
III	T1D	NO	Type 1 diabetes induction (30 mg/kg streptozocin, high dose)
IV	NO	YES	Control/Reference, UV Irradiation
V	T2D	YES	Type 2 diabetes induction (20 mg/kg streptozocin, low dose), UV Irradiation
VI	T1D	YES	Type 1 diabetes induction (30 mg/kg streptozocin, high dose), UV Irradiation

**Table 2 cancers-16-01507-t002:** The raw values of glucose measurements in the experimental mouse model (Legend: I: no treatment, no diabetes induction, no irradiation, II: Type 2 diabetes induction (low dose), no irradiation, III: Type 1 diabetes induction (high dose), no irradiation, IV: No diabetes induction, irradiation, V: Type 2 diabetic diabetes induction, irradiation, VI: Type 1 diabetes induction, irradiation). NaN: Not a Number (no measure was taken of reference mice).

	Glucose in mg/dL (Mean ± SD)							
Group		Month 0	1st Month	2nd Month	3rd Month	4th Month	5th Month	6th Month
	I	121.14 ± 2 8.57	NaN	NaN	NaN	NaN	NaN	119.43 ± 15.33
	II	118.00 ± 16.86	173.00 ± 43.29	200.83 ± 29.96	190.17 ± 25.25	160.33 ± 37.15	192.17 ± 47.88	192.83 ± 73.20
	III	128.17 ± 14.15	304.67 ± 111.39	487.67 ± 40.35	506.00 ± 58.16	543.00 ± 46.77	594.00 ± 08.04	578.25 ± 26.54
	IV	111.14 ± 19.97	NaN	NaN	NaN	NaN	NaN	108.00 ± 18.89
	V	120.17 ± 23.49	154.00 ± 34.79	179.67 ± 37.67	137.83 ± 20.20	145.50 ± 35.12	156.50 ± 15.19	114.00 ± 18.48
	VI	127.29 ± 32.70	344.14 ± 119.02	529.71 ± 58.68	547.43 ± 44.22	572.00 ± 47.61	575.67 ± 38.72	550.00 ± 70.71

**Table 3 cancers-16-01507-t003:** TEWL measurements in the experimental mouse model (Legend: I: No treatment, no diabetes induction, no irradiation, II: Type 2 diabetes induction, no irradiation, III: Type 1 diabetes induction, no irradiation, IV: No diabetes induction, irradiation, V: Type 2 diabetes induction, irradiation, VI: Type 1 diabetes induction, irradiation).

Transepidermal Water Loss (TEWL) in gr/m^2^/h (Mean ± SD)								
Group		0 Month	1st Month	2nd Month	3rd Month	4th Month	5th Month	6th Month
	I	8.73 ± 01.30	12.46 ± 03.92	10.71 ± 01.41	9.14 ± 00.69	8.93 ± 00.89	8.49 ± 01.12	9.33 ± 01.04
	II	8.84 ± 01.10	10.46 ± 02.50	12.00 ± 02.65	9.42 ± 00.49	8.58 ± 01.02	8.75 ± 01.60	10.13 ± 01.03
	III	8.41 ± 01.67	11.60 ± 02.89	11.40 ± 02.40	6.43 ± 01.37	6.50 ± 01.89	7.00 ± 01.54	7.00 ± 03.46
	IV	7.27 ± 01.10	16.36 ± 06.73	23.21 ± 04.79	22.00 ± 03.83	21.07 ± 03.14	27.00 ± 05.60	42.86 ± 08.38
	V	9.48 ± 01.42	16.30 ± 04.45	26.08 ± 03.80	24.67 ± 07.00	23.17 ± 03.06	31.17 ± 01.83	51.00 ± 06.81
	VI	8.84 ± 01.21	17.36 ± 04.46	19.14 ± 02.79	12.57 ± 00.98	11.14 ± 01.95	11.25 ± 01.37	11.00 ± 04.36

**Table 4 cancers-16-01507-t004:** The raw values of skin hydration measurements in the experimental mouse model (Legend: I: No treatment, no diabetes induction, no irradiation, II: Type 2 diabetes induction, no irradiation, III: Type 1 diabetes induction, no irradiation, IV: No diabetes induction, irradiation, V: Type 2 diabetes induction, irradiation, VI: Type 1 diabetes induction, irradiation).

Hydration in Arbitrary Units (Mean ± SD)								
Group		Month 0	1st Month	2nd Month	3rd Month	4th Month	5th Month	6th Month
	I	77.57 ± 04.08	72.43 ± 01.40	81.14 ± 04.53	77.43 ± 03.91	78.43 ± 04.16	73.14 ± 02.73	77.43 ± 03.74
	II	75.57 ± 04.83	72.43 ± 03.21	80.50 ± 02.35	77.50 ± 08.04	75.50 ± 06.47	73.83 ± 07.19	76.33 ± 06.41
	III	81.14 ± 07.78	68.86 ± 05.34	74.71 ± 04.61	63.14 ± 10.64	65.14 ± 11.61	61.40 ± 06.84	65.00 ± 07.00
	IV	79.71 ± 05.62	65.71 ± 06.16	71.00 ± 02.77	64.71 ± 03.50	61.57 ± 05.13	54.57 ± 07.11	52.43 ± 07.57
	V	78.83 ± 04.58	62.50 ± 03.51	68.50 ± 03.39	63.50 ± 05.79	63.67 ± 04.13	55.83 ± 07.60	53.67 ± 06.38
	VI	80.14 ± 03.29	64.43 ± 07.48	61.29 ± 06.45	54.71 ± 05.65	48.86 ± 07.20	38.14 ± 04.18	36.33 ± 00.58

**Table 5 cancers-16-01507-t005:** The raw values of skin layer thickness measurements in the experimental mouse model (Legend: I: No treatment, no diabetes induction, no irradiation, II: Type 2 diabetes induction, no irradiation, III: Type 1 diabetes induction, no irradiation, IV: No diabetes induction, irradiation, V: Type 2 diabetes induction, irradiation, VI: Type 1 diabetes induction, irradiation).

Skin Layer Thickness (mm) (Mean ± SD)								
		0 Month	1st Month	2nd Month	3rd Month	4th Month	5th Month	6th Month
Group	I	0.84 ± 00.08	0.84 ± 00.10	0.97 ± 00.13	0.94 ± 00.07	0.99 ± 00.10	1.00 ± 00.06	1.00 ± 00.06
	II	0.87 ± 00.04	0.87 ± 00.06	0.98 ± 00.05	0.99 ± 00.04	1.03 ± 00.05	1.06 ± 00.08	1.03 ± 00.03
	III	0.83 ± 00.13	0.79 ± 00.06	0.79 ± 00.13	0.74 ± 00.14	0.71 ± 00.13	0.70 ± 00.14	0.68 ± 00.15
	IV	0.85 ± 00.09	1.06 ± 00.18	1.49 ± 00.18	1.65 ± 00.17	.70 ± 00.14	1.86 ± 00.39	2.41 ± 00.16
	V	0.86 ± 00.10	1.05 ± 00.22	1.52 ± 00.12	1.57 ± 00.18	1.58 ± 00.21	1.74 ± 00.18	2.07 ± 00.52
	VI	0.86 ± 00.06	1.13 ± 00.09	1.34 ± 00.08	1.27 ± 00.06	1.16 ± 00.06	1.22 ± 00.10	1.32 ± 00.23

**Table 6 cancers-16-01507-t006:** Body weight measurements in the experimental mouse model (Legend: I: No treatment, no diabetes induction, no irradiation, II: Type 2 diabetes induction, no irradiation, III: Type 1 diabetes induction, no irradiation, IV: No diabetes induction, irradiation, V: Type 2 diabetes induction, irradiation, VI: Type 1 diabetes induction, irradiation).

Body Weight (gr) (Mean ± SD)								
		0 Month	1st Month	2nd Month	3rd Month	4th Month	5th Month	6th Month
Group	I	31.41 ± 04.24	33.09 ± 02.66	34.39 ± 05.58	37.99 ± 04.90	39.27 ± 04.57	42.00 ± 03.28	43.86 ± 03.44
	II	31.79 ± 02.08	34.16 ± 02.00	35.30 ± 01.84	38.57 ± 02.61	38.73 ± 02.81	38.67 ± 03.53	42.55 ± 02.65
	III	29.47 ± 03.80	30.29 ± 03.06	27.67 ± 04.64	29.36 ± 02.29	29.53 ± 02.65	30.44 ± 02.44	29.43 ± 04.38
	IV	31.23 ± 05.84	34.67 ± 06.71	33.40 ± 06.51	36.73 ± 05.53	37.17 ± 05.57	39.41 ± 06.38	39.89 ± 06.83
	V	30.48 ± 05.57	33.03 ± 05.35	32.35 ± 03.82	35.08 ± 05.44	35.03 ± 05.11	37.00 ± 05.22	37.42 ± 05.46
	VI	31.17 ± 03.60	33.30 ± 03.84	28.11 ± 00.56	29.36 ± 01.60	29.14 ± 00.97	28.03 ± 00.69	24.40 ± 03.03

**Table 7 cancers-16-01507-t007:** The raw values of skin elasticity measurements in the experimental mouse model (Legend: I: No treatment, no diabetes induction, no irradiation, II: Type 2 diabetes induction, no irradiation, III: Type 1 diabetes induction, no irradiation, IV: No diabetes induction, irradiation, V: Type 2 diabetes induction, irradiation, VI: Type 1 Diabetes induction, irradiation).

Skin Elasticity (Arbitrary Units) (Mean ± SD)					
		0 Month	1st Month	3rd Month	5th Month
Group	I	0.22 ± 00.12	0.14 ± 00.03	0.13 ± 00.03	0.19 ± 00.04
	II	0.15 ± 00.05	0.22 ± 00.09	0.16 ± 00.08	0.18 ± 00.04
	III	0.19 ± 00.10	0.20 ± 00.10	0.19 ± 00.03	0.24 ± 00.15
	IV	0.13 ± 00.03	0.14 ± 00.03	0.09 ± 00.01	0.08 ± 00.03
	V	0.19 ± 00.07	0.13 ± 00.03	0.11 ± 00.02	0.08 ± 00.02
	VI	0.15 ± 00.03	0.12 ± 00.01	0.11 ± 00.01	0.12 ± 00.02

**Table 8 cancers-16-01507-t008:** The raw values of sebum, melanin, and stripped keratinocyte measurements in the experimental mouse model (Legend: I: No treatment, no diabetes induction, no irradiation, II: Type 2 diabetes induction, no irradiation, III: Type 1 diabetes induction, no irradiation, IV: No diabetes induction, irradiation, V: Type 2 diabetes induction, irradiation, VI: Type 1 diabetes induction, irradiation).

Sebum, Melanin and Keratinocytes (Mean ± SD)				
		Sebum	Melanin	Keratinocytes
		5th Month	6th Month	6th Month
Group	I	4.14 ± 01.57	154.71 ± 37.50	4.29 ± 00.76
	II	4.33 ± 01.86	114.00 ± 11.87	4.00 ± 01.10
	III	3.00 ± 00.71	126.50 ± 26.84	6.17 ± 02.23
	IV	6.57 ± 02.30	690.00 ± 83.38	9.71 ± 01.60
	V	3.17 ± 01.47	570.17 ± 91.95	9.67 ± 02.66
	VI	2.67 ± 00.82	509.33 ± 91.43	12.67 ± 03.67

**Table 9 cancers-16-01507-t009:** The raw values of glutathione, uric acid and ROS measurements in the experimental mouse model (Legend: I: No treatment, no diabetes induction, no irradiation, II: Type 2 diabetes induction, no irradiation, III: Type 1 diabetes induction, no irradiation, IV: No diabetes induction, irradiation, V: Type 2 diabetes induction, irradiation, VI: Type 1 diabetes induction, irradiation).

		Glutathione (nmol/mg) (Mean ± SD)		Uric Acid (nmol/mg) (Mean ± SD)		Oxidative Stress (Fluorescence) (Mean ± SD)
		3rd Month	5th Month	3rd Month	5th Month	5th Month
Group	I	2.86 ± 00.78	2.62 ± 00.31	1.94 ± 00.96	2.33 ± 01.90	20,735.71 ± 438.64
	II	0.00 ± 00.00	0.00 ± 00.00	0.00 ± 00.00	0.00 ± 00.00	20,951.00 ± 287.05
	III	2.14 ± 00.44	1.90 ± 00.86	1.86 ± 00.32	3.46 ± 01.29	20,773.60 ± 448.94
	IV	1.54 ± 00.35	0.97 ± 00.17	3.45 ± 01.62	3.65 ± 01.15	21,493.14 ± 461.08
	V	0.00 ± 00.00	0.00 ± 00.00	0.00 ± 00.00	0.00 ± 00.00	21,351.75 ± 198.56
	VI	1.28 ± 00.22	0.80 ± 00.14	3.67 ± 01.33	2.34 ± 00.90	21,087.33 ± 437.39

## Data Availability

The datasets supporting the results of the current study are available from the corresponding author on reasonable request.

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
