# Peer review of "Type I Diabetes Mellitus Suppresses Experimental Skin Carcinogenesis"

_cancers, 2024, doi:10.3390/cancers16081507_

Round 1
Reviewer 1 Report
Comments and Suggestions for Authors
This paper involves a study utilizing male SKH-HR2 mice, aged 2 months to study the effects of UV radiation on a how type 1 and type 2 diabetic skin responds to ultraviolet (UV) radiation compared to normal skin. The findings of this research indicate that type 1 diabetic skin shows reduced sensitivity in developing squamous cell carcinoma and nevi. The paper contains interesting data; however, it is difficult to read some of the tables and figures. I recommend that the authors revise the manuscript to facilitate the reader’s ability to follow the details of the study and its results.
General Comments
The authors report a large amount of data that is hard to follow due to the following reasons. (A) the tables have run on rows of numbers that are hard to follow; (B) figure legends don’t really explain what is in the figures. It would be helpful in deciphering what is important in the images to provide more information; (c) details of gross animal images need to be described in more detail in the figure legend; and (D) histopathological images are very small and lack arrows to important details in the figures.
Specific Comments:
Line 128. Was the light intensity measured before or after the light was filtered through appropriate filters?
Line 130. Define what the minimal erythemal dose is?
Line 175 The skin elasticity was measured using CUTOMETER. The authors repeat a misnomer. Since skin is viscoelastic, they are measuring skin displacement under a suction not the elastic properties of skin.
Line 188 We have evaluated the skin thickness with a digital caliper (Powerfix Profi, Milomex Ltd, UK). Skin thickness can only be accurately measured using OCT or ultrasound imaging. Since the skin is viscoelastic the thickness will depend on rate of pinching and force applied.
Table 2. The raw values of glucose measurements in the experimental mouse model. Table 2 is difficult to read and should be revised. The data under group III is unreadable. What does NaN mean?
Figures 2A, 3A and 2C, 3C are difficult to read. The symbols overlap and are difficult to understand.
Tables 3, 4, 5, 6 are difficult to read. The numbers run together! Why are there two zeros in front of the decimal point in Table 7. One zero is a significant figure before the decimal point.
Figure 14. Photographic evaluation of UV irradiation effects on the experimental model. The Figure legend and photos are difficult to read and understand. What is the purpose of showing all the figures?
Figure 15. Histological evaluations of the experimental groups. The images are too small to view. Arrows are needed to augment the information in the figure legend.
Summary
The manuscript could be greatly improved by adding additional information and details to the figures. The tables are difficult to read and should be revised.
Comments on the Quality of English LanguageThe English language usage and writing style is good.
Author Response
Dear Professor Rallis,
Thank you again for your manuscript submission:
Manuscript ID: cancers-2924664
Type of manuscript: Article
Title: TYPE I DIABETES MELLITUS SUPPRESSES EXPERI-MENTAL SKIN CARCINOGENESIS
Authors: Maria Giakoumaki, George Lambrou, Dimitrios Vlachodimitropoulos, Anna Tagka, Andreas Vitsos, Maria Kyriazi, Aggeliki Dimakopoulou, Vasiliki Anagnostou, Marina Karasmani, Heleni Deli, Andreas Grigoropoulos, Evangelos Karalis, Michail Christou Rallis *, Homer Black
Received: 4 Mar 2024
Your manuscript has now been reviewed by experts in the field. Please find your manuscript with the referee reports at this link: https://susy.mdpi.com/user/manuscripts/resubmit/8dc16a761be2a79d5d7f7b932b46d4bb
Please revise the manuscript according to the referees' comments and upload the revised file within 10 days.
Please use the version of your manuscript found at the above link for your revisions.
(I) Please change references style. Please include the first ten authors'
names before "et al". Details can be found at: https://www.mdpi.com/journal/cancers/instructions
(II) Please check that all references are relevant to the contents of the manuscript.
(III) Any revisions to the manuscript should be highlighted, such that any changes can be easily reviewed by editors and reviewers.
(IV) Please provide a cover letter to explain, point by point, the details of the revisions to the manuscript and your responses to the referees’
comments.
(V) If the reviewer(s) recommended references, please critically analyze them to ensure that their inclusion would enhance your manuscript. If you believe these references are unnecessary, you should not include them.
(VI) If you found it impossible to address certain comments in the review reports, please include an explanation in your appeal.
(VII) The revised version will be sent to the editors and reviewers.
If one of the referees has suggested that your manuscript should undergo extensive English revisions, please address this issue during revision. We propose that you use one of the editing services listed at https://www.mdpi.com/authors/english or have your manuscript checked by a colleague fluent in English writing.
Please do not hesitate to contact us if you have any questions regarding the revision of your manuscript or if you need more time. We look forward to hearing from you soon.
Kind regards,
Ms. Kiran Yu
Assistant Editor
E-Mail: kiran.yu@mdpi.com
Reviewer #1
Open Review
(X) I would not like to sign my review report
( ) I would like to sign my review report
Quality of English Language
( ) I am not qualified to assess the quality of English in this paper
( ) English very difficult to understand/incomprehensible
( ) Extensive editing of English language required
( ) Moderate editing of English language required
(X) Minor editing of English language required
( ) English language fine. No issues detected
Yes |
Can be improved |
Must be improved |
Not applicable |
|
Does the introduction provide sufficient background and include all relevant references? |
(X) |
( ) |
( ) |
( ) |
Are all the cited references relevant to the research? |
(X) |
( ) |
( ) |
( ) |
Is the research design appropriate? |
( ) |
(X) |
( ) |
( ) |
Are the methods adequately described? |
( ) |
(X) |
( ) |
( ) |
Are the results clearly presented? |
( ) |
(X) |
( ) |
( ) |
Are the conclusions supported by the results? |
( ) |
(X) |
( ) |
( ) |
We would like to thank the Editor in chief, Editor and the reviewers for carefully reviewing the manuscript. We much appreciate your comments and suggestions. We have revised the manuscript accordingly. The comments have been addressed accordingly and highlighted markedly throughout the manuscript. In addition, Figures have been improved and numbers have been corrected when necessary and manuscript have been revised for typos and minor errors.
- Comments and Suggestions for Authors
This paper involves a study utilizing male SKH-HR2 mice, aged 2 months to study the effects of UV radiation on and how type 1 and type 2 diabetic skin responds to ultraviolet (UV) radiation compared to normal skin. The findings of this research indicate that type 1 diabetic skin shows reduced sensitivity in developing squamous cell carcinoma and nevi. The paper contains interesting data;
Response: We thank the reviewer for the insightful and constructive comments. We have addressed all the issues raised by the reviewer and all changes have been distinctively presented in the text.
- However, it is difficult to read some of the tables and figures. I recommend that the authors revise the manuscript to facilitate the reader’s ability to follow the details of the study and its results.
Response: All tables and figures have been substantially re-formatted in order to be clearer to the reader.
- General Comments
- The authors report a large amount of data that is hard to follow due to the following reasons. (A) the tables have run on rows of numbers that are hard to follow;
Response: All tables have been substantially re-formatted and re-designed, as suggested.
- (b) Figure legends don’t really explain what is in the figures. It would be helpful in deciphering what is important in the images to provide more information;
Response: Respective figures have been substantially re-formatted and re-designed, as suggested.
- (c) Details of gross animal images need to be described in more detail in the figure legend;
Response: Descriptions have been amended as suggested (please refer to p. 18, lines 544-548).
- (D) Histopathological images are very small and lack arrows to important details in the figures.
Response: Histopathological images were amended as suggested. Please refer to the response below.
- Specific Comments
- Line 128. Was the light intensity measured before or after the light was filtered through appropriate filters?
Response: The measurements were conducted immediately following the filters to accurately assess the UV dose administered to the animals. The manuscript has been accordingly updated to reflect this adjustment (please refer to p. 3, line 131).
- Line 130. Define what the minimal erythemal dose is?
Response: We thank the reviewer for the suggestion. MED definition was added (please refer to p. 3, lines 134-136)
- Line 175 the skin elasticity was measured using CUTOMETER. The authors repeat a misnomer. Since skin is viscoelastic, they are measuring skin displacement under a suction not the elastic properties of skin.
Response: Thank you for your insightful comments regarding the use of the Cutometer to measure skin elasticity in our manuscript. We appreciate the opportunity to clarify this aspect of our study and to reinforce the appropriateness of our methods grounded in established precedents. You rightly point out that skin is viscoelastic and that what the Cutometer measures is skin displacement under suction, not the elastic properties of the skin. This enhanced understanding is crucial and is, in fact, a principle that underpins the methodology of using Cutometer in dermatological research. The Cutometer has been designed to quantify the viscoelastic properties of the skin by measuring its ability to resist from deformation, which is. It's important to note that the term “elasticity” as used in the context of Cutometer measurements and reported in a vast body of literature (https://www.courage-khazaka.de/images/Downloads/Studien/Studies_Cutometer.pdf), encompasses both elastic and viscoelastic properties of the skin. This terminology has become standard in dermatological research because it effectively captures the functional outcomes of interest. Furthermore, the widespread acceptance and use of the Cutometer as an “elasticity meter” in dermatological research are supported by its consistent, reproducible measurements across numerous studies. It is recognized as a valuable tool for assessing changes in skin properties in response to various interventions, conditions, or treatments. To support this point, we have provided citation in our manuscript that validate the use of the Cutometer in measuring skin elasticity and its relevance in dermatology. We understand the importance of precision in our language and methodology and have taken care to ensure our manuscript explains the capabilities of the Cutometer. We believe that our use of the Cutometer and the terminology associated with its measurements are in line with current dermatological research practices. However, to address your concern and to avoid any potential confusion, we have added a clarification in the methods section of our manuscript. This clarification explains the viscoelastic nature of skin and how the Cutometer's measurements relate to the comprehensive assessment of skin's elastic properties. We hope this response adequately addresses your concerns and justifies our methodology. We are grateful for the chance to improve our manuscript with your valuable feedback (please refer to p. 5, lines 192-196).
- Line 188 we have evaluated the skin thickness with a digital caliper (Powerfix Profi, Milomex Ltd, UK). Skin thickness can only be accurately measured using OCT or ultrasound imaging. Since the skin is viscoelastic the thickness will depend on rate of pinching and force applied.
Response: Skin fold measurement is a technique primarily used to estimate body fat percentage in humans by measuring the thickness of skin folds at various body sites. While it is less common to apply this method directly to measuring skin thickness in mice due to the different tissue composition and scale, the principle behind the technique can offer some value when interpreted with caution. Indeed, direct measurement of skin thickness in mice, typically through histological examination or using tools like calipers directly on excised skin, provides more precise and accurate data. These methods directly measure the skin without the confounding effects of underlying fat or muscle tissue, which can vary significantly between individuals and strains of mice. However, in contexts where non-invasive, cost effective and less technically demanding methods are required, skin fold measurement can serve as a proxy to infer changes in skin characteristics over time or between experimental groups. It is noteworthy that this method is widely used as skin thickness assessment in the literature. (Rather IA, Bajpai VK, Huh YS, Han YK, Bhat EA, Lim J, Paek WK, Park YH. Probiotic Lactobacillus sakei proBio-65 Extract Ameliorates the Severity of Imiquimod Induced Psoriasis-Like Skin Inflammation in a Mouse Model. Front Microbiol. 2018 May 17;9:1021. Doi: 10.3389/fmicb.2018.01021. PMID: 29867905; PMCID: PMC5968580.) In conclusion, while skin fold measurement is not a direct method for assessing skin thickness in mice, with careful application and interpretation, it can provide insights into changes in skin and subcutaneous tissue properties in a non-invasive manner. Your point underscores the importance of most accurate method description and thus the relevant parts of manuscript have been changed (please refer to p. 5, lines 199-202).
- Table 2. The raw values of glucose measurements in the experimental mouse model. Table 2 is difficult to read and should be revised. The data under group III is unreadable. What does NaN mean?
Response: Table 2 was revised accordingly. We have re-formatted Table 2, to be easier to read. NaN, indicates Not a Number (or no measurement taken). We have added the abbreviation in the table legend (please refer to Table 2).
- Figures 2A, 3A and 2C, 3C are difficult to read. The symbols overlap and are difficult to understand.
Response: Figures 2A, 3A, 2C, 3C have been re-formatted as suggested.
- Tables 3, 4, 5, 6 are difficult to read. The numbers run together! Why are there two zeros in front of the decimal point in Table 7. One zero is a significant figure before the decimal point.
Response: Tables have been re-formatted as suggested. All corrections have been performed as suggested.
- Figure 14. Photographic evaluation of UV irradiation effects on the experimental model. The Figure legend and photos are difficult to read and understand. What is the purpose of showing all the figures?
Response: We thank the reviewer for his insightful comment. In this figure our intention was to present the evolution of skin aberrations in three indicative experiments, in such a way that it would be easier for the reader to understand how skin is affected after UV exposure and under the presence of DM. the figure presents the effects of UV radiation and the combined effect of DM in indicative mice experiments. The text and the figure legends were amended as suggested (please refer to p. 18, lines 548-563 and figure 14 legend)
- Figure 15. Histological evaluations of the experimental groups. The images are too small to view. Arrows are needed to augment the information in the figure legend.
Response: Histological images were amended as suggested (please refer to p. 21, lines 570-577 and figure 15 legend)
We would like heartfully to thank the reviewer for his efforts and time to help ameliorate the manuscript.
Permit me please to express our gratitude.
Reviewer 2 Report
Comments and Suggestions for Authors
The paper investigates the ambitious hypothesis that type 1 diabetes appears to be a protective factor for the occurrence of UV-related skin cancers.
The study is conducted in a mouse model; the authors select six groups of mice, each consisting of seven mice.
It is shown that the UV-exposed group consisting of mice with type 1 diabetes mellitus is less susceptible to skin tumors and precancerosis than the other exposed groups consisting of mice with type 2 or wild-type diabetes mellitus.
Several aspects are investigated: skin hydration, atrophy, dermoscopy and histology of the different mouse groups.
The materials and methods are well explained. Some tables need improvement in readability of numbers; other images are clear.
English is fluent.
The discussion is also well conducted and the conclusions are in line with the study.
All in all, this is a good, very audacious work that opens up interesting prospects for the future, both from a pathogenetic and clinical-interventional perspective.
Author Response
Dear Professor Rallis,
Thank you again for your manuscript submission:
Manuscript ID: cancers-2924664
Type of manuscript: Article
Title: TYPE I DIABETES MELLITUS SUPPRESSES EXPERI-MENTAL SKIN CARCINOGENESIS
Authors: Maria Giakoumaki, George Lambrou, Dimitrios Vlachodimitropoulos, Anna Tagka, Andreas Vitsos, Maria Kyriazi, Aggeliki Dimakopoulou, Vasiliki Anagnostou, Marina Karasmani, Heleni Deli, Andreas Grigoropoulos, Evangelos Karalis, Michail Christou Rallis *, Homer Black
Received: 4 Mar 2024
Your manuscript has now been reviewed by experts in the field. Please find your manuscript with the referee reports at this link: https://susy.mdpi.com/user/manuscripts/resubmit/8dc16a761be2a79d5d7f7b932b46d4bb
Please revise the manuscript according to the referees' comments and upload the revised file within 10 days.
Please use the version of your manuscript found at the above link for your revisions.
(I) Please change references style. Please include the first ten authors'
names before "et al". Details can be found at: https://www.mdpi.com/journal/cancers/instructions
(II) Please check that all references are relevant to the contents of the manuscript.
(III) Any revisions to the manuscript should be highlighted, such that any changes can be easily reviewed by editors and reviewers.
(IV) Please provide a cover letter to explain, point by point, the details of the revisions to the manuscript and your responses to the referees’
comments.
(V) If the reviewer(s) recommended references, please critically analyze them to ensure that their inclusion would enhance your manuscript. If you believe these references are unnecessary, you should not include them.
(VI) If you found it impossible to address certain comments in the review reports, please include an explanation in your appeal.
(VII) The revised version will be sent to the editors and reviewers.
If one of the referees has suggested that your manuscript should undergo extensive English revisions, please address this issue during revision. We propose that you use one of the editing services listed at https://www.mdpi.com/authors/english or have your manuscript checked by a colleague fluent in English writing.
Please do not hesitate to contact us if you have any questions regarding the revision of your manuscript or if you need more time. We look forward to hearing from you soon.
Kind regards,
Ms. Kiran Yu
Assistant Editor
E-Mail: kiran.yu@mdpi.com
Reviewer #2
Open Review
(X) I would not like to sign my review report
( ) I would like to sign my review report
Quality of English Language
( ) I am not qualified to assess the quality of English in this pape
( ) English very difficult to understand/incomprehensible
( ) Extensive editing of English language required
( ) Moderate editing of English language required
( ) Minor editing of English language required
(X) English language fine. No issues detected
Yes |
Can be improved |
Must be improved |
Not applicable |
|
Does the introduction provide sufficient background and include all relevant references? |
(X) |
( ) |
( ) |
( ) |
Are all the cited references relevant to the research? |
(X) |
( ) |
( ) |
( ) |
Is the research design appropriate? |
(X) |
( ) |
( ) |
( ) |
Are the methods adequately described? |
(X) |
( ) |
( ) |
( ) |
Are the results clearly presented? |
(X) |
( ) |
( ) |
( ) |
Are the conclusions supported by the results? |
(X) |
( ) |
( ) |
( ) |
Comments and Suggestions for Authors
The paper investigates the ambitious hypothesis that type 1 diabetes appears to be a protective factor for the occurrence of UV-related skin cancers. The study is conducted in a mouse model; the authors select six groups of mice, each consisting of seven mice. It is shown that the UV-exposed group consisting of mice with type 1 diabetes mellitus is less susceptible to skin tumors and precancerosis than the other exposed groups consisting of mice with type 2 or wild-type diabetes mellitus. Several aspects are investigated: skin hydration, atrophy, dermoscopy and histology of the different mouse groups. The materials and methods are well explained. Some tables need improvement in readability of numbers; other images are clear. English is fluent. The discussion is also well conducted and the conclusions are in line with the study. All in all, this is a good, very audacious work that opens up interesting prospects for the future, both from a pathogenetic and clinical-interventional perspective.
Response: We heartfully thank the reviewer for his efforts and comments. We are really pleased that the reviewer identified the key concepts of the work. All tables and figures have been re-formatted in οrder to be clearer to the reader. Finally, we feel the need to express our gratitude for the kindness and big heart of the reviewer.